

**Analysis of the streamflow extremes and long term water balance**
**in Liguria Region of Italy using a cloud permitting grid spacing**
**reanalysis dataset**
Francesco Silvestro[1],Antonio Parodi[1], Lorenzo Campo[1] , Luca Ferraris[1]
[1]{CIMA Research Foundation, Savona, Italy}
Correspondence to: francesco.silvestro@cimafoundation.org
**Abstract**
Characterizing the hydrometeorological extremes, both in terms of rainfall and streamflow,
as well as the estimation of long term water balance indicators are essential issues for the
flood alert and water management services which are in charged to provide  environmental
monitoring. In recent years simulations carried out with meteorological models  are getting
available at increasing spatial and temporal resolutions (both historical reanalysis and near
real-time hindcast studies); these meteorological data sets can thus be used as input in
distributed hydrological models to drive long-period hydrological reanalysis. In this work we
adopted a high resolution meteorological reanalysis dataset that covers the whole Europe
territory for the period between 1979 and 2008 , with 4 km grid spacing and 3 hours of time
resolution. This reanalysis dataset is used together with a rainfall downscaling algorithm and a
rainfall bias correction technique in order to produce input to a continuous and distributed
hydrological model; the resulting modelling chain allows to produce long time series of
distributed hydrological variables, inter alia streamflows and evapotranspiration, in the
Liguria Region of Italy territory, located in the Northern part of Italy, and among the western
Mediterranean areas mostly impacted by severe hydro-meteorological events.
The observations available from the local rain gauges network were compared with the
rainfall estimated by the dataset , and then used to perform a bias correction with the aim  of
matching the observed climatology. An analysis of the annual maxima discharges derived by





simulated streamflow timeseries was carried out, by comparing them with observed discharge
where available and using as benchmark a regional statistical analyses elsewhere. Eventually
an investigation of long term water balance was done by comparing simulated runoff
coefficients with available estimations based on observations.
The study highlights both limits and potentialities of the considered framework as a
methodological approach to undertake hydrological studies in any point of a considered study
area mainly characterized by a collection of small basins, thus allowing to overcome the
limits of observations which are punctual and in some cases not fully reliable.

## 1  Introduction

The estimation of magnitude of discharge and the probability of occurrence of a certain
streamflow is an important task for a number of purposes: risk assessment, design of
structural protections against flooding, civil protection aims, settings of thresholds in early
warning systems.
Standard approaches for facing this issue are using streamflow observations to carry out a
statistical analysis on a particular closure section (Kottegoda and Rosso, 1997), but this is not
always possible because of lack of observations; to solve this problem a frequency
regionalization approach can be used (De Michele and Rosso, 2002) even using both
observations and streamflow derived by hydrological modelling (Boni et al., 2007).
On the other hand even studies and methodologies regarding the management of water
resources and drought have an important role especially in the perspective of possible future
changes in climate and water needs (Calanca et al., 2006; Fu et al., 2007; Döll and Müller,
2012). In this case the analysis of long-term water balance components is of primary
importance, and the evaluation of total runoff and evapotranspiration becomes crucial.
In the last decades the use of Regional Reanalysis for hydrological purposes and to study
basins behaviour in different hydrological regimes is began quite frequent, even because the



reliability of meteorological variables estimation derived by the reanalysis is constantly
improving together with the increasing of reanalysis spatial grid spacing and time resolution.
Among many others, Choi et al. (2008) investigated the applicability of temperature and
precipitation data from the North American Regional Reanalysis (NARR, about 32 km grid
spacing) for hydrological modelling of selected watersheds in northern Manitoba, while
Bastola and Misra (2013) used reanalysis products as surrogates for large-scale observations
and showed that they are superior in simulating hydrological response in respect to other four
considered meteorological datasets. Furthermore, Krog et al. (2015) used ECMWF interim
reanalysis (ERA-Interim, about 70 km grid spacing) as input to a hydrological model in order
to better understand the processes that drive the hydrological response of one of the largest
rivers in Patagonia, similarly Nkiaka et al. (2017) investigates the potential of using global
reanalysis datasets as input for hydrological modelling in the data-scarce Sudan-Sahel region.
In this work a very high-resolution ($\Delta t=3$ h, $\Delta x=4$ km) regional dynamical downscaling of
historical climate scenarios is used as input to a continuous distributed hydrological model
producing high-resolution ($\Delta t=1$ h, $\Delta x=< 500$ m) 30 years length modelled variables history
for a reference Mediterranean region. On one side the distributed nature of the state and
output variables allowed to investigate the possibility of using this kind of modelling chain for
extreme streamflow statistical analysis (e.g. distribution of annual discharge maxima) and
long term water balance (e.g. long term runoff coefficient) in a fully distributed perspective.
On the other side the high spatial grid spacing and time resolution of forcings, together with
the use of a rainfall downscaling model, allowed to explore the use of such high resolution
reanalysis in regions characterized by the presence of small hydrological watersheds in areas
characterized by very complex topography.
The study shows the capabilities and the limits of the considered modelling chain to
reproduce low frequency streamflow and long term water balance, in order to evaluate





possible applications, as an alternative to the use of observations, in a scarce data environment
or in those cases where the spatial distribution of hydro-meteorological processes results to be
essential.
The manuscript is organized as follows: section 2 describes the study area, hydro-
meteorological data set and models, section 3 shows the results, and in section 4   the
discussion and conclusions are reported.

## 2    Materials and Methods

### 2.1    Study Area and Case study

Liguria Region is located in northern Italy (Figure 1), it is characterized by basins which have
steep valleys due to their proximity to the Apennines and Alps and drainage area in the range
$10^1$ to $10^3$   $km^2$,. The response time to precipitation pulses of these basins is short, ranging
between 0.5 to 10 hours. The maximum elevations are around 2500 m, with very steep slopes,
and  a large part of the ground surface is covered with forest or other types of vegetation with
the catchments mouth is often in correspondence of towns and cities.
A quite dense meteorological stations network monitors the territory of Liguria, it is , named
OMIRL – "Osservatorio Meteo-Idrologico della Regione Liguria" and it is managed by the
Environmental Agency of Liguria Region (ARPAL). This monitoring network yields
raingauge measurements with timesteps smaller than 1 hour (e.g. 5-10 minutes) for more than
150 gauges over the region; the density is averagely 1 raingauge/40 $km^2$. Temperature,
radiation, wind, air humidity gauges are part of the system, even though their density is lower
than the rain gauges one.
For a subset of raingauge stations, historical validated data are available from 1978 to 2010;
in fact ARPAL built a database useful for climatic and statistical analysis which is freely
available   on   web   (http://www.arpal.gov.it/homepage/meteo/analisi-climatologiche/atlante-



climatico-della-liguria.html). These data were used in the presented study for rainfall bias
correction.
Furthermore annual discharge maxima time series longer than 30 yrs are available for a set of
level gauges (Figure 1).
**2.2  EXPRESS-Hydro reanalysis**
The fifth climate model intercomparison project (Coupled Model Intercomparison Project
Phase 5, CMIP5) produced a rich portfolio of global scale present and future climate
numerical simulations in different emission scenarios, using hydrostatic global climate
models (GCM) available only at quite coarse spatial resolutions. Intense precipitation events
over complex topography areas such as Europe are generally not accurately represented by the
aforementioned GCMs. Consequently the CORDEX (COordinated Regional climate
Downscaling Experiment) initiative is aiming at producing regional climate change
projections worldwide for input into impact, adaptation and disaster risk reduction studies
using regional climate models (RCM) at fine spatio-temporal resolution and forced by
different GCMs from the CMIP5 archive. Along these lines Kotlarski et al. (2014) confirmed,
with simulations on grid-resolutions up to about 12 km (0.11°), the capability of RCMs to
correctly reproduce the main features of the European climate for the period 1989-2008,
moreover they also exhibit relevant modelling errors: as an example precipitation biases are in
the ±40% range while seasonally and regionally averaged temperature biases are generally
smaller than 1.5 °C. Building on these findings, Pieri et al. (2015) moved one step further, in
the framework of the EXtreme PREcipitation and Hydrological climate Scenario Simulations
(EXPRESS-Hydro) project, dynamically downscaling, at very high spatio-temporal
resolution, the historical climate scenarios generated by the ERA-Interim reanalysis using the
state-of-the-art non-hydrostatic Weather Research and Forecasting (WRF) regional climate
model.



Indeed, Pieri et al. (2015) performed for the first time long climate simulations (1979-2008)
over the European domain (Inner European Region, IER, Figure 2) at a very fine cloud-
permitting resolution of about 4 km (0.037°) with explicitly resolved convection and a sharp
representation of orography. Pieri et al. (2015) assessed the WRF regional climate model
performances to reproduce observed precipitation extremes, over Europe and with a particular
focus on the Greater Alpine Region, by comparing the simulations results with available
gridded observational data sets such as the high-resolution Alpine precipitation grid dataset
(APGD) developed by MeteoSwiss in the framework of the European Reanalysis and
Observations for Monitoring (EURO4M) collaborative project (Isotta et al. 2013). Overall
Pieri et al. (2015) results showed that increased grid spacing together with the use of
explicitly resolved convection allows to achieve a better modelling of the precipitation field
spatio-temporal distribution, reducing significantly the overestimation of precipitation
(around 5% on annual average over the European domain with respect to E-OBS
observational data) and thus better reproducing the distribution and the statistics of the rainfall
rate, particularly over the Alps and Apennines area. More recent studies confirmed the need
for high-resolution dynamical downscaling for extreme weather impact studies in regions
with complex terrain  (Pontoppidan et al. 2017, Schwitalla et al. 2017).
**2.3    Bias correction of rainfall fields (B.C.)**
Before being used as input for the hydrological simulations, the Pieri et al (2015) reanalysis
rainfall dataset was compared with the precipitation data obtained by the Liguria raingauge
network record (Atlante Climatico Liguria, http://www.arpal.gov.it/homepage/meteo/analisi-
climatologiche/atlante-climatico-della-liguria.htm). The observed dataset is constituted by
validated time series of about 80 raingauges homogeneously distributed on the Liguria Region





territory: the time series cover the whole 1979-2008 period of the EXPRESS-Hydro dataset
(even if not for all gauges) at hourly timestep.
In order to provide to the hydrological model Continuum the most correct input data possible,
the Pieri et al (2015) data were BIAS-corrected by employing the observed data to assure an
accurate reproduction of the rainfall climatology of the area in terms of monthly cumulate.
Different methods are available in literature to perform a BIAS correction (hereafter B.C.) on
different variables (rainfall, temperature, etc., Fang et al. 2015): in this work a CDF-matching
approach was selected (Fang et al. 2015). In order to preserve both the seasonality and the
inter-annual variability of the observations, the correction was based on the monthly
cumulates, that were computed for both the Express-Hydro dataset (thus producing N x 12
maps representing the average cumulate rainfall map for each month, N number of years of
the Express-Hydro dataset) and the observed dataset (producing time series of $N_{obs}$ x 12
values representing the average cumulate rainfall height for each month and for each of the
available raingauges, $N_{obs}$ number of years of the observed dataset).
To allow a direct comparison between the observed data and the modeled dataset, the monthly
cumulated data from the raingauges were previously interpolated on the Express-Hydro
spatial grid, by using a kriging technique with a Spherical variogram. No regression with
other spatialized variables (e.g. elevation) was employed because previous tests showed that
no significant correlation were present.
For each cell i, the empirical CDF (Cumulative Distribution Function) of observed and
modeled values were computed. At the purpose of minimizing distortions of the information,
these CDFs were computed separately for each month of the year (January, February, etc.).
In the CDF-matching process the observations CDF was imposed to the Expresss-Hydro time
series of a given cell i in order to obtain the corrected time series of the monthly cumulate:


$$PM'_{i,m} = F_{OSS,i}^{-1}\left(F_{MOD,i}\left(PM_{i,m}\right)\right)$$

where $PM$ is the Express-Hydro monthly cumulate rainfall, $PM'$ is the bias-corrected monthly
cumulate rainfall, $m$ is the index of the month of the original series, $F_{MOD,i}$ and $F_{OSS,i}$ are
respectively the CDF of the modeled and observed monthly rainfalls in the cell $i$.

5       Given these corrected monthly time series, the single instantaneous value of rainfall $p$

(3-hours cumulate) was corrected as follows:

$$p'_{i,t} = p_{i,t}\,\frac{PM'_{i,m}}{PM_{i,m}}$$

where:
-   $p_{i,t}$ is the 3-hours cumulate rainfall modeled in the cell $i$ at instant $t$
-   $p'_{i,t}$ is the bias-corrected 3-hours cumulate rainfall modeled in the cell $i$ at instant $t$
-   $PM_{i,m}$ is the monthly cumulate rainfall modeled in the cell $i$ in month $m$ (in which the
instant $t$ falls)
-   $PM'_{i,m}$ is the monthly cumulate rainfall modeled in the cell $i$ in month $m$ (in which the
instant $t$ falls) corrected with the CDF-matching
The described procedure allowed obtaining a 3-hours maps dataset in which the model bias
was eliminated by keeping the characteristics of the model in terms of seasonality and inter-
annual variability. It is important to highlight that the CDF-matching procedure did not alter
the presence of possible temporal trends, at both domain and cell spatial scale.
**2.4    Downscaling the precipitation with RainFARM Model**
RainFARM (Rebora et al. 2006a, 2006b) is a mathematical model able to downscale rainfall
fields that can be exploited for generating rainfall scenarios consistent with large scale
forecasts done by the Numerical Weather Prediction Systems (NWPS) as in Laiolo et al.
(2013) and/or by expert forecasters (Silvestro and Rebora, 2014). RainFARM accounts for the
variability of precipitation fields at small spatial and time scales (e.g. L < 1 km, t < hour),
preserving the precipitation volume at those scales where quantitative precipitation forecasts
are considered reliable ($L_r$, $t_r$). RainFARM is able, on one side to preserve spatial and time



patterns at $L_r$, $t_r$, on the other side to produce small-scale structures of rainfall which are
consistent with detailed remote sensor observations as meteorological-radar estimation.
In the model the spatial-temporal Fourier spectrum of the precipitation field is estimated
using rainfall fields predicted by a meteorological model and it is mathematically described as
follows:

$$\left|\hat{g}(k_x, k_y, \omega)\right|^2 \propto (k_x^2 + k_y^2)^{-\alpha/2} \omega^{-\beta} \qquad (3)$$

$k_x$ and $k_y$ are the x and y spatial wavenumbers, $\omega$ the temporal wavenumber (frequency),
while $\alpha$ and $\beta$ are two parameters that are calibrated fitting the power spectrum of rainfall
derived by a NWPS on the frequencies that correspond to the spatial-time scales $L_r$, and $t_r$.
.Extending the spectrum defined by equation (3) to the larger wave numbers/frequencies it is
possible to generate a spatial-time rainfall field at a high resolution (Rebora et al. 2006b).
Since the Fourier phases related with the power spectrum (3) are randomly generated before
the backwards transformation in real space, RainFARM can give as output an ensemble of
equi-probable high-resolution fields that remain coherent at large scale with the rainfall
forecast done by NWPS. RainFARM was designed to be used for flood forecast systems
implemented on small and medium sized basins (drainage area $< 10^3$-$10^4$), anyway in this
study is used to downscale EXPRESS-Hydro reanalysis; the reliable spatial and time scales
are assumed as the nominal grid spacing and temporal resolution of EXPRESS-Hydro
precipitation (4 km and 3 hours, Hardenberg et al. 2015 and Pieri et al. 2015). The
downscaling algorithm is not used in probabilistic configuration, but to build a possible
rainfall time-spatial pattern at 1 km and 1 hour resolution that is compatible with the runoff
formation at small scales, since most of the catchments in the proposed the study area the





catchments have reduced dimensions (often <100-200 km$^2$) with response time in the order of
1-6 hours.
**2.5    The hydrological model: Continuum**
The hydrological model used in this study is *Continuum* (Silvestro et al. 2013), it is a
distributed and continuous model that relies on a space-filling approach, and uses a
consolidated way for the identification of the drainage network components (Giannoni et al.,
2005). All of the main hydrological processes are mathematically described in a distributed
way in Continuum but it was designed to be a balance solution between complex physically
based models which describe the physical phenomena with high detail often introducing
complex parameterization and models with a empirical approach, easy to implement but far
from reality, (Silvestro et al., 2015). Continuum can be implemented in different contexts
even on data scarce environments. The basin or domain of interest is represented through a
regular grid, derived by a Digital Elevation Model (DEM) while the flow directions are
defined with an algorithm that calculates the directions of maximum slope using the DEM.
An algorithm classifies each cell of the drainage network as  hillslope or channel flow
depending on the main flow regime. A morphologic filter defined by the expression $AS^k = C$
is used to distinguish  between hillslopes and channels ; A is the drainage area upstream of
each cell [L$^2$], S is the local slope [-], k and C are constants related to the geomorphology of
the catchment (Giannoni et al., 2000). The surface flow scheme treats differently channel and
hillslope flows. The overland flow (hillslopes) is described by a linear reservoir
schematization, while an approach derived by kinematic wave (Wooding, 1965; Todini and
Ciarapica, 2001) is used for modeling the channel flow.
Subsurface flows and infiltration and are modelled using a methodology based on a
adaptation of the Horton equations (Bauer, 1974; Disikin and Nazimov, 1997); it accounts for





soil moisture evolution even in conditions of intermittent and low-intensity rainfall as
described in Gabellani et al., (2008).
Interception of vegetation is schematized with a reservoir that has a retention capacity $S_v$; this
latter isestimated using static informative layers of vegetation type or Leaf Area Index data;
finally the flow in deep soil and the water table evolution are modeled with a distributed
linear reservoir schematization and a simplified version of Darcy equation.
The energy balance uses the "force restore equation" (Dickinson, 1988) that allows to
explicitly model the soil surface temperature.
The parameters that require calibration in Continuum model are six, they are often estimated
at basin or sub-basin scale: two for the surface flow ($u_h$ and $u_c$), two for the sub-surface flow
($c_t$ and $c_f$) and two for deep flow and watertable ($V_{Wmax}$ and $R_f$) processes.
The parameter $u_h$ affects those hydrograph components which are related to fast surface flow
as well as $u_c$ but the impact of this latter depends on the length of the channeled paths. $c_f$ is
related to saturated hydraulic conductivity and controls the rate of subsurface flow rate (i.e.,
it). $c_t$ identifies the part of water volume in the soil that can be extracted through
evapotranspiration only and is thus related to the soil field capacity, while Both $c_t$ and $c_f$
regulate the dynamics of saturation of the root-zone. The two parameters $V_{Wmax}$ and $R_f$ rule
the flow in the deep soil and the dynamic of watertable, they impact on recession curves and
have certain influence on flood hydrographs, only when catchments of large drainage area are
considered (Silvestro et al., 2013).
In the presented work, Continuum was implemented with a time resolution of 60 min and
with a spatial resolution of 0.005 deg (about 480 m). The Shuttle Radar Topographic Mission
(SRTM) DEM was used.





It was possible to calibrate the model for 11 sections where streamflow observations are
available; ground stations measurements were interpolated on a regular grid of 1 km
resolution by using Kriging method and used as input to the model. 01/01/2013-31/12/2014
was used as validation period, and the following skill scores were employed to measure the
model performances.
The Nash Sutcliffe (NS) coefficient (Nash and Sutcliffe, 1970):
$$NS = 1 - \sum_{t=1}^{t\,max} \frac{\left(Q_m(t) - Q_o(t)\right)^2}{\left(Q_m(t) - \overline{Q_o}\right)^2} \qquad (4)$$
Where $Q_m(t)$ and $Q_o(t)$ are the modelled and observed streamflows at time t. $\overline{Q_o}$ is the mean
observed streamflow.
Relative Error of High Flows (REHF)
$$REHF = \frac{1}{t\,max}\left[\sum_{t=1}^{t\,max} \frac{\left|Q_m(t) - Q_o(t)\right|}{Q_o(t)}\right]_{Q>Qth} \qquad (5)$$
Where $Q_{th}$ is chosen as the 99 percentile of the observed hydrograph along the calibration
period.
NS and REHF were combined in a multi-objective function to carry out the calibration using
the approach proposed in Madsen (2000).
The values of the skill scores were calculated for the validation period and results to be
satisfactory; they are reported in Table 1.
In those basins where it was not possible to make the calibration, average values of the
parameters obtained by the calibration process are assumed.





In order to have reduced warm up impacts on 1979 simulation a first run was done and the
state variables simulated on the 31 of December of every year (from 1979 to 2008) were used
to estimate an average initial condition to be used in the final simulation.
Once the run of hydrological model for the period 1979-2008 was done, we had available a
streamflow time series for each pixel of the calculation domain, ideally it is like having a sort
of gauge every $\Delta x$ along the stream network. Given the spatial resolution of the hydrological
model we discarded the analysis for basins with area smaller than $A_{th}=15$ km$^2$, since we retain
that the surface water motion processes are not modeled with a sufficient detail; $A_{th}=15$ km$^2$
means a number of pixel around 60-70 to represent the catchment and at least some pixels
classified as channel pixel (Giannoni et al., 2005).
**3   Results**
**3.1   Precipitation analysis**
The comparison between precipitation climatology over Liguria from observational data and
Pieri et al (2015) results has been undertaken at the annual and seasonal scales.
Pieri et al. (2015), using EURO4M-APGD reference observational dataset (Isotta et al. 2013 ,
with about 50 daily raingauge stations over Liguria), already showed an overall
underestimation of the WRF rainfall depths on the annual basis in Liguria, more evident on
the eastern side of the region (Fig4, panels e-f of Pieri et al. 2015), with differences in the
range between -2 and -1 mm on the coastal eastern Liguria portion and between -1 and 0
mm/day on the eastern Appennine side.
The same analysis has been repeated in this study, using about 80 raingauge stations over
Liguria: the total annual rainfall depth results are largely confirmed both on eastern and
weastern Liguria sides (Figure 3).




Concerning the seasonal temporal scales Pieri et al (2015) results tend to underestimate
average observed rainfall depths during DJF (Figure 4) on eastern Liguria portion (between -1
and 0 mm/d) while overall it overestimates them on central-western Liguria (0-1 mm/d).
Similar argument holds also for the MAM period (Figure 5), even if the WRF overestimation
are tends to get larger on western Liguria side. During the JJA period (Figure 6), instead, Pieri
et al (2015) results exhibit an overestimation of observed average rainfall depths between 0
and 1 mm/d over the western and eastern Liguria sides, while it is increasing to 1-2 mm/day
over central Liguria. The understimation gets worse over eastern Liguria during SON (Figure
7), with values between -2 and -1 mm/d over the inland portions and up to -3 and -2 mm/d
over the coastal one. Conversely on the rest of the region the understimation falls around -1
and 0 mm/d.
Figure 8 shows the box-plot of monthly precipitation averaged at regional scale for both
EXPRESS-Hydro and Observations. For each month we have in fact two time series,
observed and modeled, of rainfall accumulation values obtained averaging the maps    The
comparison evidences that EXPRESS-Hydro reproduces quite well the variability along the
30 years but often underestimates the rainfall amount, this is particularly evident in January,
September and October.
**3.2   Distribution of the annual discharge maxima**
The results of the modeling chain were firstly compared with observations using a typical
station wise comparison approach: 15 gauge stations with at least 30 years of annual
discharge maxima (hereafter ADM) were identified along the Ligurian territory from the
eastern side to the western one.





Indeed, it was not possible to ensure a perfect overlapping between the simulation period and
the observed data availability, in many cases observed data are not temporally continuous and
they may cover longer periods (in some case 50-60 years) with large time windows of missing
data. However on the basis of the conclusions drawn in the framework of the Atlante
climatico della Liguria, major climate change related trends for temperature, precipitation,
and so forth are not apparent, thus supporting the database usage despite these data gaps.
The comparison between observed and reanalysis driven annual discharge maxima (ADM) is
firstly based on the analysis of the respective cumulative density function distributions.
The reanalysis driven ADM were fitted with a General Extreme Value (GEV) distribution
(see e.g., Hoskin and Wallis, 1993; Piras et al., 2015) that represents a good compromise
between flexibility and robustness. Other works based extreme statistical analysis on the two
components extreme value (TCEV) model (Rossi et al., 1984), anyway we decided to use
GEV since has a smaller number of parameters and it was successfully applied for a wide
number of applications (CIMA research foundation, 2015)  In figures 9 to 11 a series of
graphs are presented where a selection of observed and reanalysis driven ADM CDF
distributions are compared, together with the GEV obtained by reanalysis driven ADM and
the corresponding 95% confidence intervals. For each station both the results obtained with
and without rainfall B.C. are reported. The comparisons correspond to six hydrological
gauging stations where reliable and long-time series of observed ADM are available.
The cases in Figure 9 show a shift of the observed distribution with respect to the modeled
one especially without B.C. Low ADM observed values lay out of the confidence intervals of
the reanalysis driven ADM GEV distribution, while the most extreme values are inside the
confidence intervals. The distributions without bias correction show underestimation of



ADM; B.C. leads to very good results for Entella closed at Panesi case and to an
overestimation on Bisagno closed at La Presa case.
Magra closed at Piccatello (Figure 10) shows an overestimation in both cases, which is
getting higher after the B.C; Argentina closed at Merelli benefits of B.C. especially regarding
the extreme ADM values.
Arroscia closed at Pogli shows an improvement of reanalysis driven ADM, once B.C. is
performed, while Nervia ADM without B.C. fit well observations and B.C. leads to an
overestimation (Figure 11).
The Kolmogorov-Smirnov test with 5% significance level was applied to all the selected
stations and corresponding results are summarized in Figure 12, in which x-axis shows the
section number, while y-axis shows the P-values, the legend reports the number of stations
that passed the test. It is interesting that BIAS correction does not allows to increase the
number of null-hypothesis (data belong to the same distribution) but even without bias
correction we have 9 stations on 15 that pass the test. In some stations B.C. worsens the
results in some other improves the results. Changing the significance level the final findings
do not change very much: with 1% 12 stations pass the test with B.C. 11 without B.C., with
10% 7 stations pass the test with B.C. 7 without B.C. . This fact could derive on how the B.C.,
that acts on the monthly volume, affects short and severe rainfall events in different parts of
the study area; these intense and short events are often the ones that in many cases cause the
ADM.
**3.3    Regional analysis of the annual discharge maxima**
In order to carry out a comparison following a distributed approach we referred to the work
done in Boni et al. (2007) which is one of the methods operationally used in Liguria Region





(by public authorities and private engineer) to estimate the ADM quantiles (Provincial
Authority of Genoa, 2001; Silvestro et al., 2012). The method was conceived and tested
especially for the Tyrrhenian catchments of Liguria Region, so the presented analysis was
carried out only for this area. The method defines a hierarchical approach based on the
analysis of the non-dimensional random variable $X_0 = X/\mu_x$, obtained by grouping together all
available data, and making them non-dimensional with respect to each local (gauging station)
sample mean, $\mu_x$, taken as the index flood for gauged sites. Index flood is estimated even
where observations were not available with support of rainfall regional frequency analysis and
rainfall-runoff modelling in order to allow quantile estimation in each point of the region.
The final result is a methodology to estimate the index flood that can be formalized as it
follows:
$$Q_{index} = f(Area, longitude)$$
While the quantile is:

$$Q(T) = K(T) \cdot Q_{index}$$

Where T is the return period and K(T) is defined by the non-dimensional regional growth
curve. Boni et al. 2007 defined a unique K(T) applicable to all the studied region.
In the case of the modelling chain analyzed in this work the reanalysis driven time series are a
large number since we can pick the 30 years long ADM series for each pixel of the model grid
where drainage area is larger than $A_{th}$. In practice the fact that we are using a distributed
hydrological model, on one side allows the index flood estimation as a simple mean of a time
series in each point of the domain, on the other side furnishes a large number of data to build
the non-dimensional regional curve.
Figure 13 shows the comparison between the non dimensional regional growth curve obtained
fitting a GEV on simulated ADM obtained with and without B.C., the observations (available
ADM on Liguria Region) and the Simulated ADM; results are quite good even if it seems that





modeling chain without B.C. leads to a small underestimation of high frequency events (low
T) and a small overestimation of low frequency events (high T) in respect to the observations.
Anyway both observed and modeled ADM lay inside the confidence intervals (95 %) for a
large part of the curve.
The main differences in the case of B.C. configuration are that observations lay always inside
the confidence intervals and there is a better matching between simulated and observed
sample curves
To compare the quantiles estimated using the modelling chain with those obtained in Boni et
al. (2007) the following ratio was considered:

$$Ratio(T) = \frac{Q(T)_{Model}}{Q(T)_{Reg}}$$

Where Model and Reg stand for modelling chain and regional analysis, T is return period, Q
is the ADM. Ideally, if modelling chain would furnish exactly the same results of the
benchmark regional analysis, Ratio(T) should be around the value 1. Ratio(T) > 1 means
overestimation in respect to the benchmark, Ratio(T)<1 means underestimation.
Figure 14 shows Ratio(T) for T=2.9 years (Index Flow) as function of the drainage area (A in
km$^2$), while Figures 15 and 16 show the maps of Ratio(T) for T=2.9 years and T=50 years.
The first consideration is that there is a relation between Ratio(T) and A, probably the chain
cannot reproduce in detail those meteorological and hydrologic processes at very small time
and spatial scales, that produces sufficient runoff for extreme flow estimation (Siccardi et al.,
2005). In fact for A < 30-50 km$^2$ the underestimation seems quite systematic even if B.C.
improves results. Ratio(T) shows a general underestimation also for A>30-50 km$^2$ but B.C.
generally leads to a better distribution between over and underestimation. Looking at Figure
15 there is a general improvement with larger values of Ratio(T) especially where the model



chain without B.C. underestimates Q(T). Ratio(T) for T= 50 years (Figure 16) has similar
pattern.
In the centre of Liguria Region Ratio(T) EXPRESS Hydro leads to results which are
comparable with findings of Boni et al. (2007) even without B.C.. Simulations with B.C.
apparently seems affected by overestimation. This could be due to different causes: i) it is
possible that EXPRESS-Hydro well reproduces the events at small time and spatial scales (3-
6 hours, 10-100 km$^2$) in that part of the region but generally underestimates monthly
cumulates, in this case B.C. could lead to streamflow overestimation; ii) hydrological model
could need a better calibration, but this is in our opinion not the case since in a calibrated
basin B.C. lead to overestimation even in the site comparison (Bisagno creek, section 3.2) iii)
we could also consider the possibility that maybe Boni et al. (2007) underestimates quantiles
in this area. It is to be noticed that overestimation for T=2.9 years is larger than that for T=50
years, this is due by the shape of the growing curve in the B.C. case
Western part of Liguria has similar behaviour even if less stressed and mainly evident for
larger basins only.
The underestimation on smaller catchments (A < 30-50 km$^2$) is presumably due to the fact
that the modelling chain cannot adequately reproduce the rainfall structures at fine spatial and
temporal scale (1 km, 1 hour or less) and the runoff processes needed to trigger such very
small catchments seems underestimated (Siccardi et al., 2005). So we can say that the
aforementioned underestimation of quantiles for very small catchments is a structural problem
of the modelling chain.
This fact is corroborated by the analysis shown in Figure (Figure 17 left panel). The Ratio(T)
averaged on the target area is plotted as a function of T for all the sections with drainage Area
> 16 km$^2$. Ratio increases with T especially for bias correction case, this means that growth
curve values (K(T)) obtained by EXPRESS-Hydro partially balance the underestimation of


average ADM (used as Index Flow) in the estimation of higher quantiles. Ratio(T=2.9 years)
changes from 0.47 to 0.71, Ratio(T=50 years) changes from 0.45 to 0.66. If we increase the
threshold area from 16 to 50 km$^2$ (Figure 17 right panel) results improve with a reduction of
the underestimation for both cases (with and without bias correction).
As already shown the general underestimation of Ratio(T) for small catchments is not
completely confirmed for all the region, in the central part of Liguria Region there is an area
where results are quite good even for basins with Area < 50 km$^2$; bias correction leads to an
overestimation of Ratio(T) in some cases. Apparently in this area EXPRESS-Hydro can
produce rainfall spatial-temporal structures able to trigger flood events compatible with the
hydro-climatology of small basins. This is also the area of the study region that previous study
demonstrated to be characterized by highest values of rainfall maxima for 1, 3, 6, 12 and
hours (Boni et al., 2008).
**3.4 Water balance and runoff coefficient**
In this paragraph some considerations about the long-term water balance are shown in order
to evaluate how the applied system can reproduce hydrological cycle and some variables
interesting for water balance and water management purposes.
To do this we estimated the distributed runoff coefficient (RC) at cell scale as:

$$RC(x,y) = \frac{Rain(x,y) - Evt(x,y)}{Rain(x,y)}$$

Where Rain(x,y) and Evt(x,y) are the total modeled rainfall and evapotranspiration in the cell
(x,y) over the 30 years of simulation: this is interesting to have an idea of the pattern of RC
and consequently of the Evapotranspiration. The maps of RC are plot in Figures 18 and 19,
together with maps of annual mean rainfall for both cases: with and without B.C..





Spatial pattern of RC is strongly correlated to spatial pattern of precipitation, and this latter is
evidently related with orography.
When a single cell has a large number of upstream cells, it tends to be frequently saturated
because of the contributions of subsurface flow of the upstream cells; as a consequence we
decided to not show the values in the cells that belong channel network as modeled by the
hydrological model (Giannoni et al., 2005), they are poorly representative; generally the
values are very low and even negative.
B.C. produces an increasing of precipitation all over the entire region and a reduction of
differences between coastal and inland area. Even the runoff coefficient increases with rainfall
B.C.
In order to estimate how the modeling chain represents the runoff coefficient at basin scale,
we considered some closure sections where runoff coefficient estimated by observations
(rainfall and streamflow) are available. They can be found in the Hydrologic Annual Survey
(http://www.arpal.gov.it/homepage/meteo/pubblicazioni/annali-idrologici.html), which is an
official document published by the Regional Agency for Environment Protection. The values
do not correspond to and cover exactly the simulation period (1979-2008) but they are often
an average of non-continuous periods since about 1940 to nowadays. Anyway they are a
possible benchmark to assess the performance of the modeling chain.
The modeled runoff coefficient of a target section s is evaluated as

$$RCs(s) = \frac{VQ(s)}{Rain(s)}$$

Where Rain(s) is the accumulated rainfall over the basin upstream the section s and VQ(s) is
the total streamflow volume passed through section s.
Results are reported in table 2. Values are compatible with the hydro-climatology of the target
area (Barazzuoli and Rigati, 2004; Provincia di Imperia, 2017) but at the same time it is





evident a general underestimation in the western part of the region (basins 4,5,6). B.C.
improves results and RCs are more similar to the benchmark.
In terms of runoff coefficient there are only small variations introducing the rainfall BIAS
Correction, this means that considering the long term water balance, the increasing or
decreasing of rainfall lead on similar percentage variation on runoff and on
evapotranspiration.
For example in the center and east of Liguria, bias correction leads to a general increasing of
precipitation and to a reduction of the orographic feature of the spatial pattern, but at the same
time the evapotranspiration increases. As shown in sections 3.2 and 3.3 the increasing of
rainfall leads to larger values of ADM, but the RCs do not change significantly.
This result could be due by the fact that even other EXPRESS-Hydro variables would need
correction, for example the solar radiation or wind which are important forcing for energy
balance (and so for long term water balance), but no reliable and dense data are available for
the entire simulation period. Another reason could be found in the hydrological model, since a
calibration more devoted to preserve long term runoff could lead to better results. In any case
we could say that generally the results are good and they evidence the potentialities of using
such modeling chain even for water balance purposes.

## 18 4    Discussion and conclusions

This work explores the possibility of using EXPRESS-Hydro, a very high-resolution regional
dynamical downscaling of ERA-Interim reanalysis with a state-of-the-art non-hydrostatic
Weather Research and Forecasting (WRF) regional climate model, for hydrological purposes
on small catchments. This was done using a subset of EXPRESS-Hydro meteorological
variables as input to a distributed continuous hydrological model to produce streamflow
simulations. The rainfall fields were downscaled from original time resolution (3 hours) to
finer one (1 hours). All the analysis was conducted with and without applying a bias



correction to rainfall fields. The study area is the Liguria Region in Italy with a particular
focus on Tyrrhenian coast.
Firstly we evaluated the performance of the presented modelling chain in reproducing
extreme streamflow statistics. This was done following two approaches: i) comparing
statistical distribution of ADM with observations in some measurement points ii) using as
benchmark the regional analysis presented in Boni et al. (2007) that allows a comparison with
a distributed perspective.
Secondly we evaluated how the modeling chain reproduces the long term water balance
analyzing the modeled runoff coefficients and using estimations based on observations as
benchmark.
The results are encouraging even if the modelling chain cannot always reproduce with high
accuracy the considered benchmarks. The ADM statistic is reproduced quite well in various
part of the target region but there are sub-regions where there is a general underestimation of
the quantiles. Rainfall B.C. leads to a general improvement reducing the general
underestimation but introducing an overestimation in some basins especially in the central
part of the region.
Comparison of modelled and observed ADM on single site shows that for a large number of
the measurement sites the time series belong to the same distribution with significance 5%;
the fitting of modelled ADM with GEV distribution are generally good and often observations
lays inside the confidence intervals at 95% of significance especially for low frequent
quantiles.
Comparison with regional analysis shows interesting results with different behaviour in
different parts of the region depending on how EXPRESS-Hydro generates the spatial-
temporal patterns of precipitation and how rainfall bias correction correct the quantitative
amounts.



Both punctual and distributed analysis evidence that there is a general underestimation for
basins with drainage area smaller than 30-50 km$^2$ but B.C. considerably corrects this
underestimation. This is probably due to structural problems of the modelling chain under the
aforementioned drainage area, for this class of basins it is necessary to further go down with
time and spatial scales in generating meteorological input, especially precipitation (Siccardi at
al., 2006; Silvestro et al., 2016), and presumably even in hydrological modelling (Yang et al.,
2001). A possibility to deal with very small basins is study a way to better exploit the
potentialities of the downscaling algorithm (RainFARM), that is here used in a deterministic
way only to generate a possible temporal-spatial pattern with 1h and 1 km spatial resolution
but maintaining the precipitation volumes and structures generated by EXPRESS-Hydro at its
resolution (3 hours, 4 km).
Runoff coefficient was used as indicator to evaluate long term water balance. The runoff
coefficient evaluated on 30 years length simulation period at cell scale, has reasonable values
for the climatology of the region (Barazzuoli and Rigati, 2004; Provincia di Imperia) and its
pattern is highly correlated with annual mean rainfall pattern. The runoff coefficient at basin
scale was compared with estimations based on observation for some measurement points, the
values are quite good as order of magnitude but generally the modelling chain underestimate
in both analyzed configurations; B.C. improves the results. This could be due by the fact that
even variables related to energy balance (for example the solar radiation and wind) modelled
by EXPRESS-Hydro probably need correction, this opportunity is not developed in the
presented work, mainly for lack of reliable and sufficiently dense data.
To summarize the results of the presented investigation, we could state that the perspective of
using the present modelling chain to produce hydro-meteorological statistical analysis in the
study area is good. The fully distributed approach allows to reproduce the hydro-climatic
characteristics and features in a continuous way along the territory. Rainfall B.C. contributes





in a relevant way to improve the results, helping the system to better model some rainfall
characteristics not completely captured even by a high-resolution meteorological reanalysis.
Very small basins (Area < 30-50 km$^2$) generally suffer of a structural underestimation only
partially corrected by B.C..
**Acknowledgements**
This work is supported by the Italian Civil Protection Department, by Environment Protection
Agency of Liguria region of Italy (ARPAL ) and by the Italian Region of Liguria.

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





1  # 5   Tables

| Basin | Section | Drainage Area [km$^2$] | NS | REHF |
|---|---|---|---|---|
| Magra | Calamazza | 1650 | 0.81 | 0.14 |
| Vara | Nasceto | 202 | 0.83 | 0.10 |
| Entella | Panesi | 364 | 0.77 | 0.18 |
| Bisagno | Passerella Firpo | 92 | 0.26 | 0.16 |
| Neva | Cisano | 123 | 0.71 | 0.25 |
| Arroscia | Pogli | 204 | 0.74 | 0.31 |
| Argentina | Merelli | 188 | 0.84 | 0.21 |
| Bormida | Murialdo | 134 | 0.35 | 0.51 |
| Bormida | Piana Crixia | 273 | 0.76 | 0.41 |
| Orba | Tiglieto | 76 | 0.88 | 0.21 |
| Aveto | Cabanne | 33 | 0.73 | 0.41 |

2      Table 1: hydrological model validation; skill score values obtained for the calibrated

3      sections





| Basin | Section | N. Progr | Area [km$^2$] | Obs. RCs. | Model RCs | Model RCs (B.C.) |
|---|---|---|---|---|---|---|
| Magra | Piccatello | 1 | 78 | 0.61 | 0.62 | 0.63 |
| Vara | Nasceto | 2 | 203 | 0.7 | 0.64 | 0.66 |
| Entella | Panesi | 3 | 364 | 0.73 | 0.64 | 0.67 |
| Neva | Cisano | 4 | 123 | 0.59 | 0.47 | 0.49 |
| Arroscia | Pogli | 5 | 202 | 0.55 | 0.48 | 0.51 |
| Argentina | Merelli | 6 | 188 | 0.65 | 0.51 | 0.53 |

3     Table 2. Runoff coefficients obtained by the modeling chain (with and without the

4     rainfall bias correction) compared to those estimated by observations.



**6  Figures**

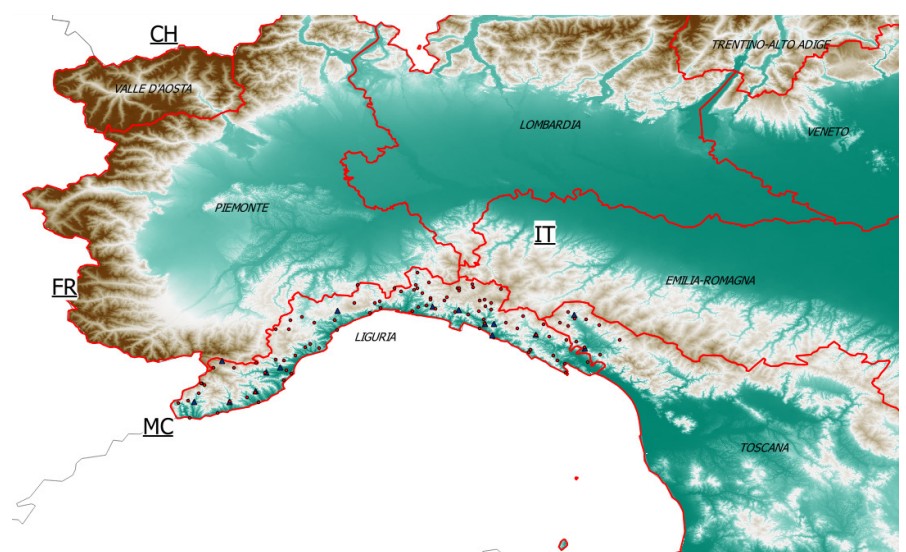

Figure 1. Study area. Red lines represent the regions of Italy, red dots represent the
meteorological gauge network of Liguria region of Italy, blue triangles are the level
gauge sections. Digital elevation model highlights the morphology of the region.



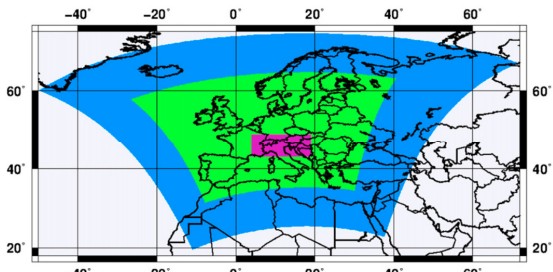

Figure 2: European domain defined in CORDEX (0.118; blue) and the IER (0.0378;
green) used for the high-resolution integration. The Great Alpine Region (GAR) used
for some of the diagnostics is displayed in purple (courtesy of Pieri et al. 2015).

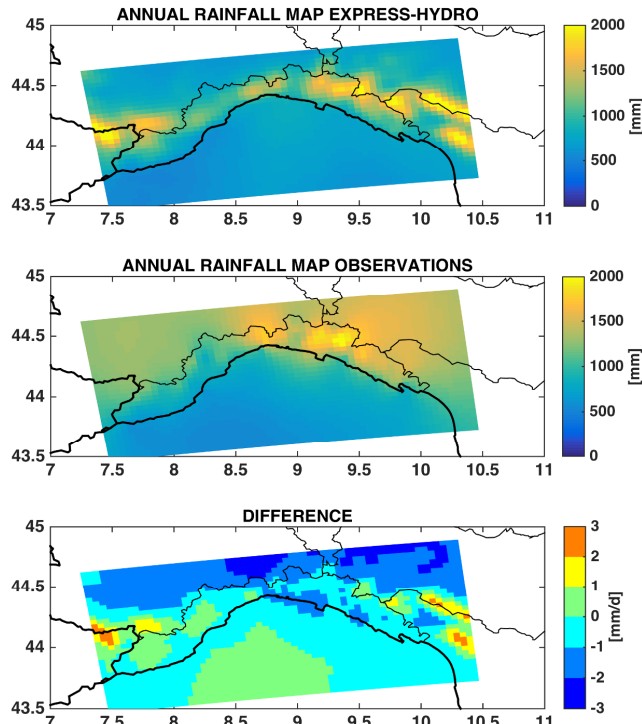

Figure 3. Upper panel shows the average annual rainfall map over Liguria area as provided by
Pieri et al (2015) results, the middle panel shows the average observed annual rainfall map,
while the bottom one shows their difference in mm/d.





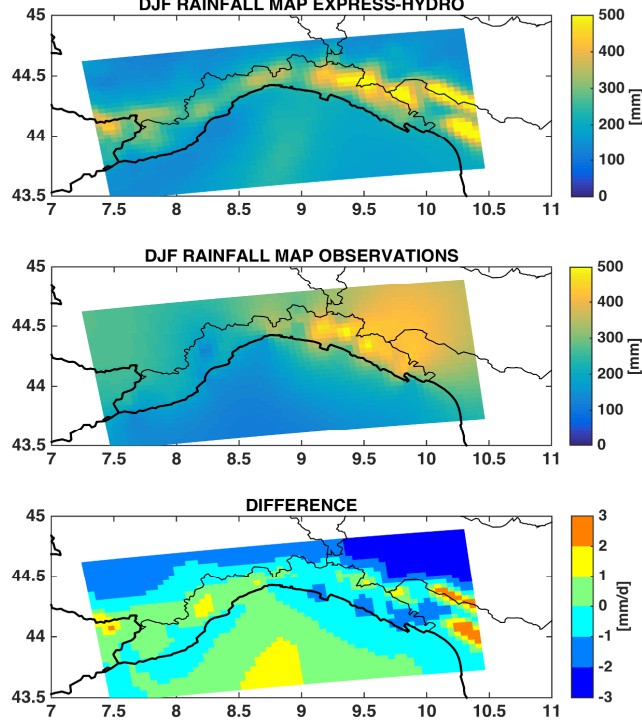

2    Figure 4. Upper panel shows the average DJF rainfall map over Liguria area as provided by

3    Pieri et al (2015) results, the middle panel shows the average observed DJF rainfall map,

4    while the bottom one shows their difference in mm/d.




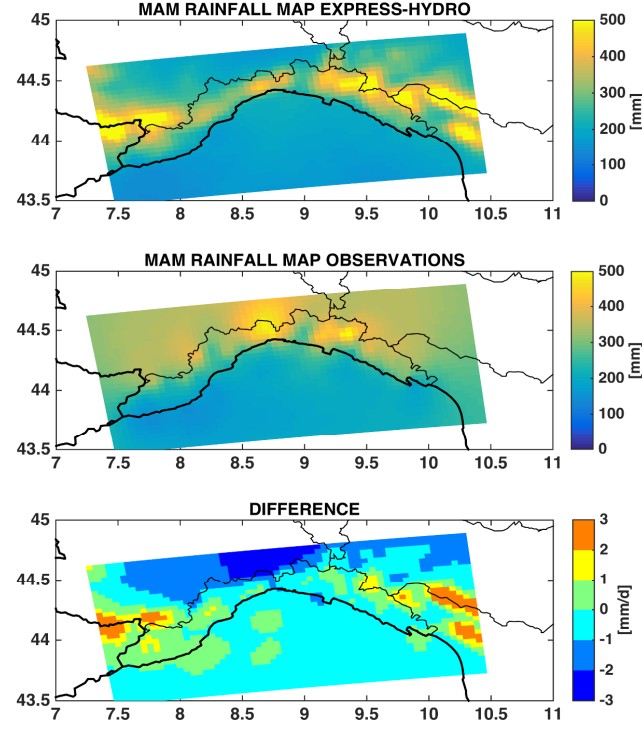

3 Figure 5. Upper panel shows the average MAM rainfall map over Liguria area as provided by

4 Pieri et al (2015) results, the middle panel shows the average observed MAM rainfall map,

5 while the bottom one shows their difference in mm/d.





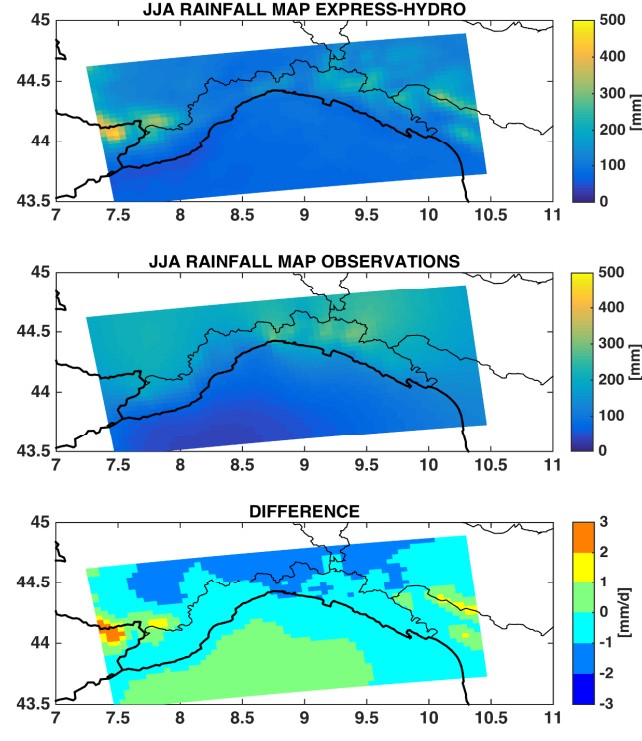

3 Figure 6. Upper panel shows the average JJA rainfall map over Liguria area as provided by

4 Pieri et al (2015) results, the middle panel shows the average observed JJA rainfall map,

5 while the bottom one shows their difference in mm/d.





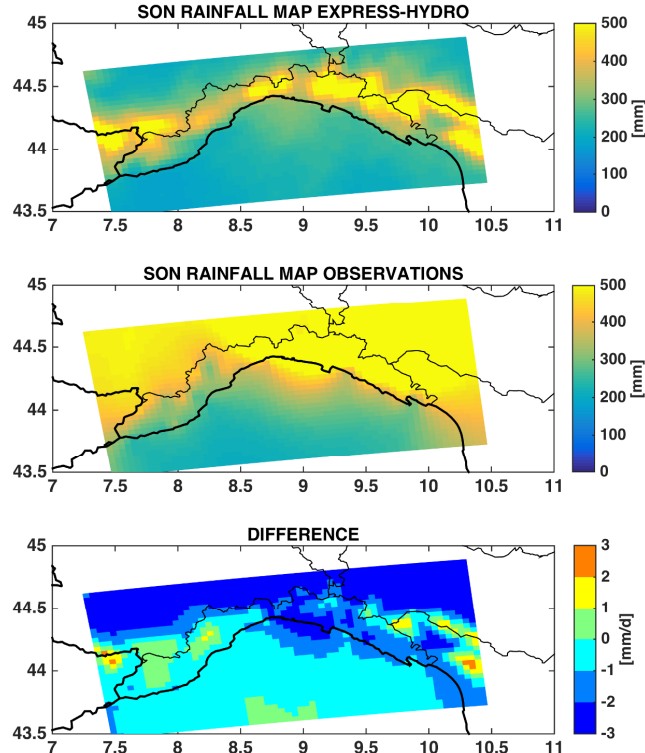

2  Figure 7. Upper panel shows the average SON rainfall map over Liguria area as provided by

3  Pieri et al (2015) results, the middle panel shows the average observed SON rainfall map,

4  while the bottom one shows their difference in mm/d.





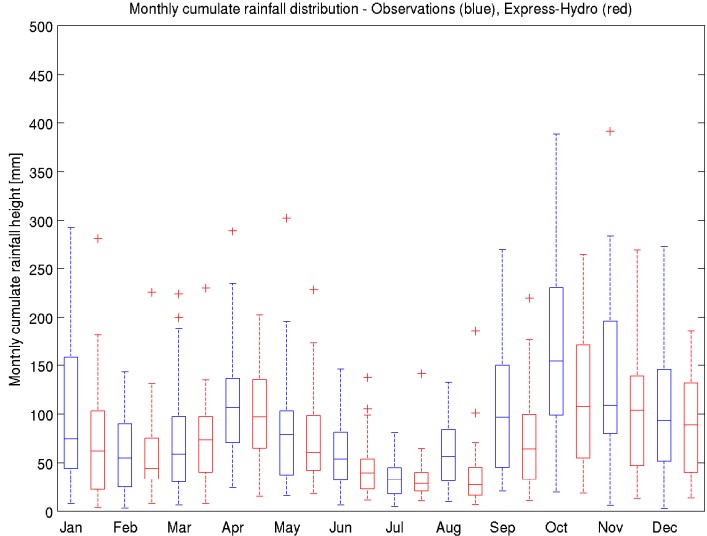

Figure 8: box plot of monthly precipitation averaged at regional scale. Blue box plots are built
with observations while red ones with EXPRESS-Hydro reanalysis




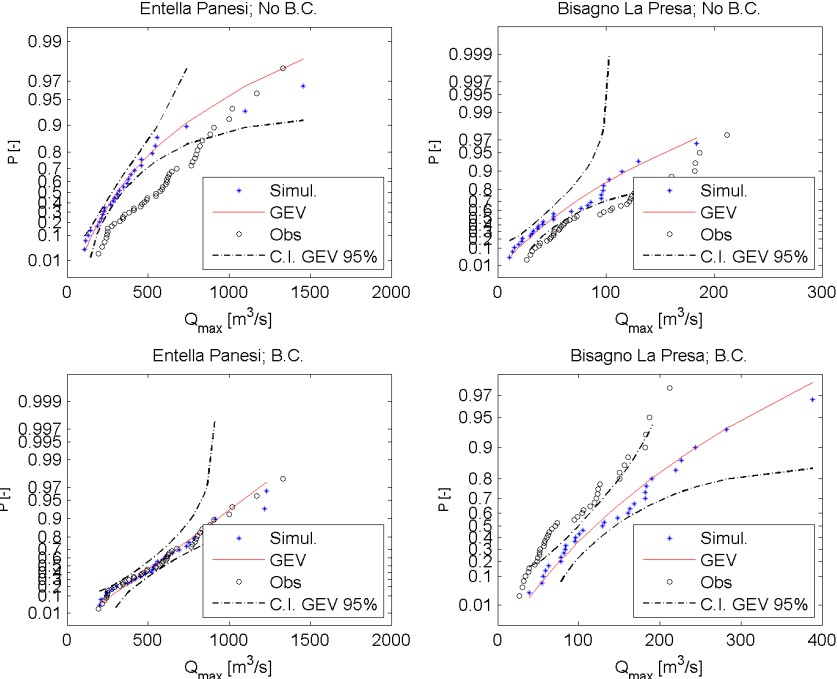

Figure 9. Distribution of ADM for Entella closed at Panesi (364 km$^2$) and Bisagno
closed at La Presa (34 km$^2$). Blue dots are the simulated ADM, black dots are observed
ADM, red line is the GEV fitted on simulated ADM while dotted lines are confidence
intervals with 95% significance. Upper panels show results without rainfall bias
correction, bottom panels show results with rainfall bias correction.



1            :

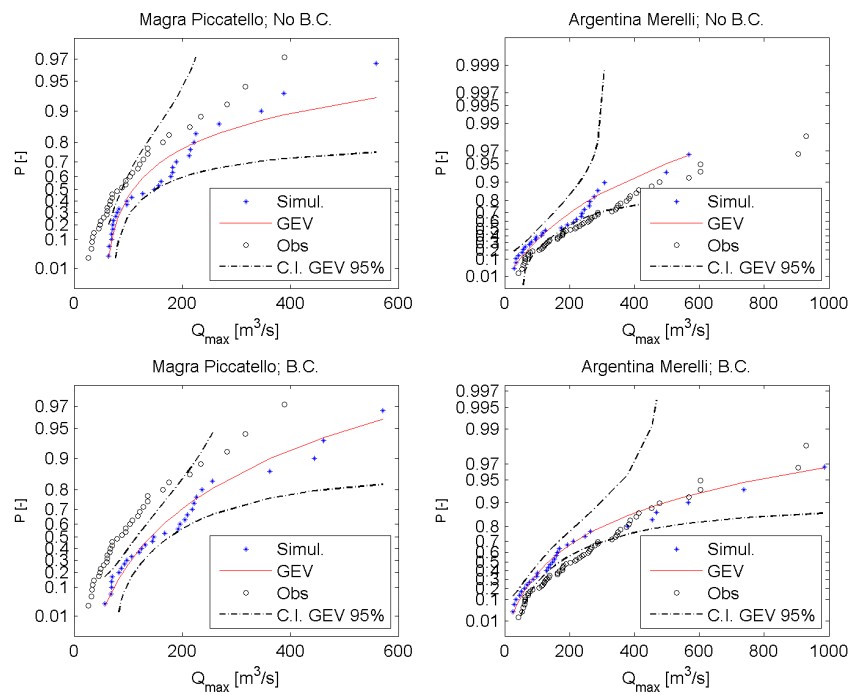

3            Figure 10. same as figure 6 but for Magra closed at Piccatello (78 km$^2$) and Argentina

4            closed at Merelli (188 km$^2$)



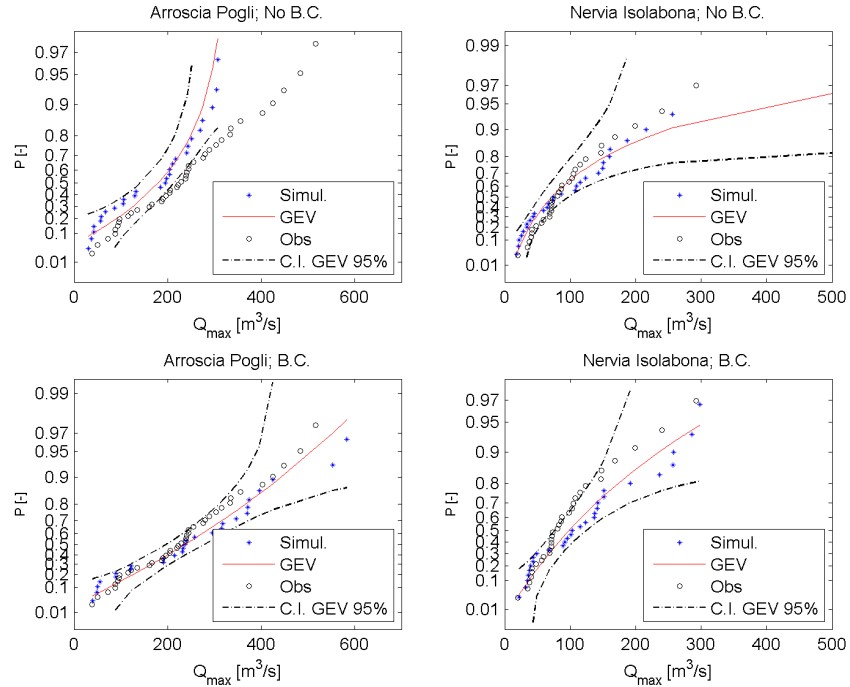

2      Figure 11: same as figure 6 but for Neva closed at Cisano (123 km²) and Nervia closed

3      at Isolabona (122 km²)

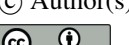



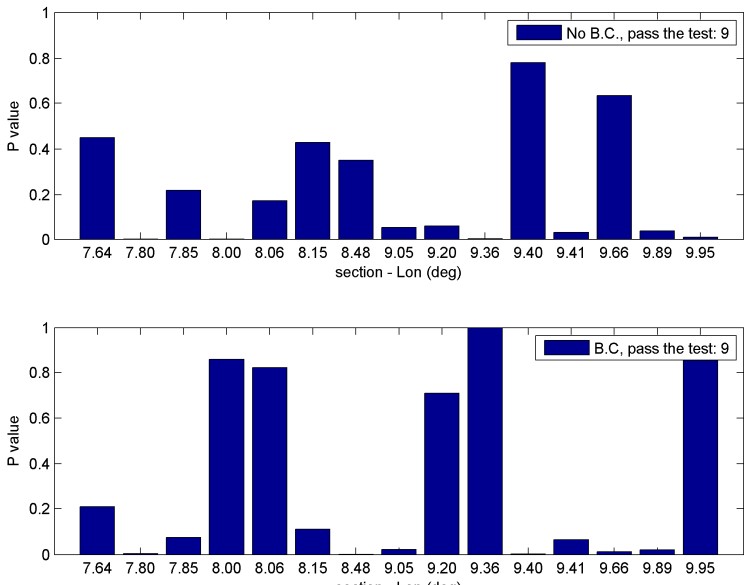

2    Figure 12: Kolmgorov Smirnov test with significance 95% done on ADM with and

3    without rainfall bias correction. On x axis the longitude of the basin centroid is reported

4    while y axis shows the P-value of the test for each section





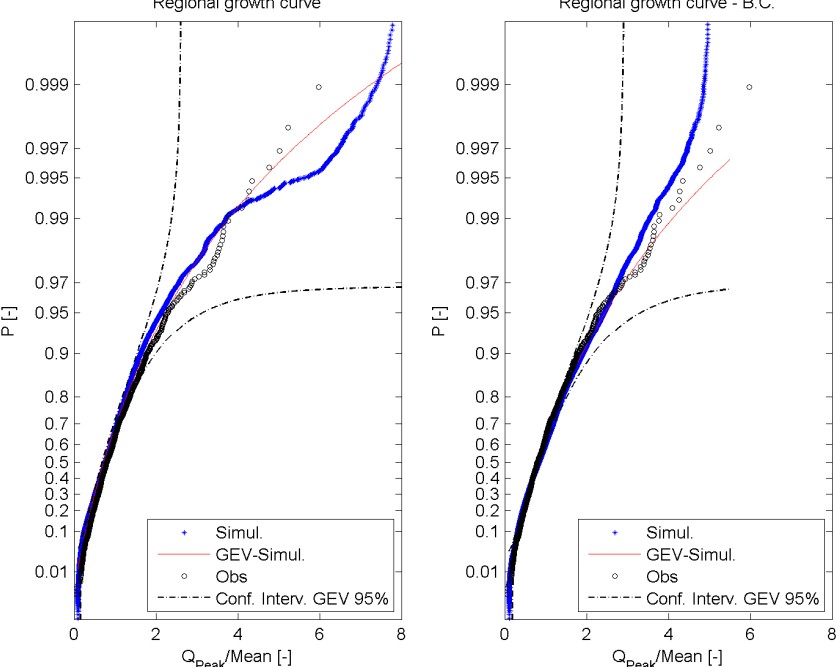

Figure 13. sample growth curve obtained by model chain (blue dots) compared with
observations (black dots). Red line is the GEV distribution fitted on modeled values
while dotted lines are the confidence intervals with significance 95%. Left panel: results
without rainfall bias correction, right panel: results with rainfall bias correction.





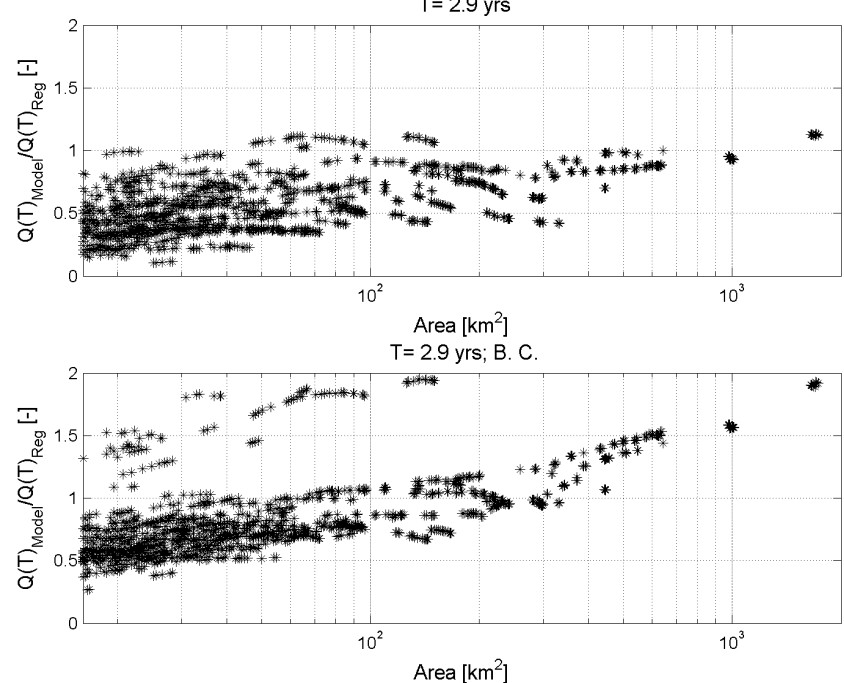

2    Figure 14. Ratio(T) as a function of drainage area. T=2.9 years which correspond to index

3    flow. Upper panel shows results without rainfall bias correction, lower panel (B.C.) with

4    rainfall bias correction.



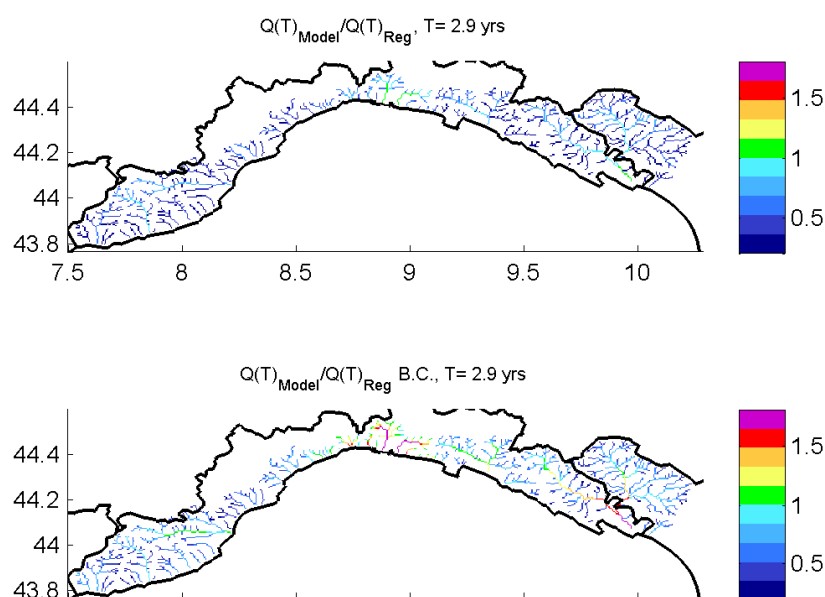

2      Figure 15. Maps of Ratio(T) for T=2.9 years. Upper panel shows results without rainfall

3      bias correction, lower panel (B.C.) with rainfall bias correction. The B.C. increases the

4      percentage of drainage network points that have values around 1.





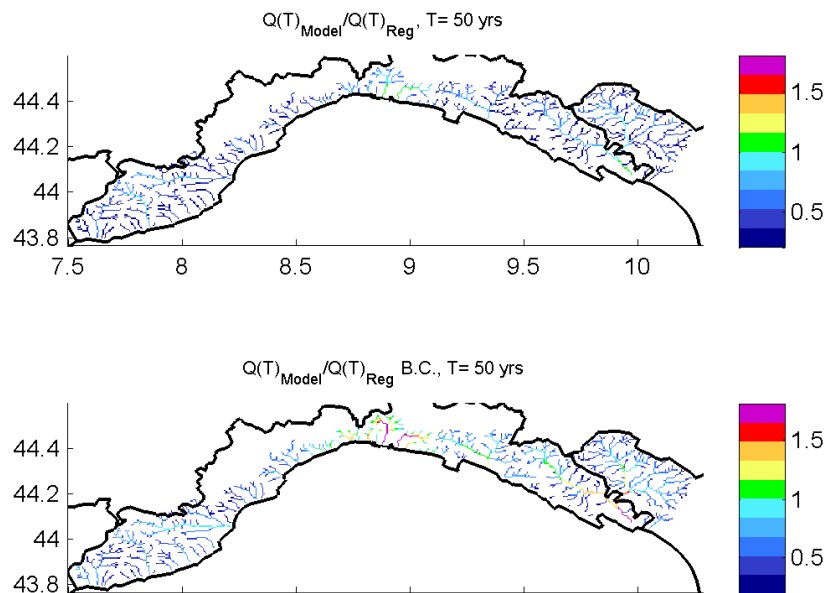

2 Figure 16. Same as figure 12 but for T=50 years.





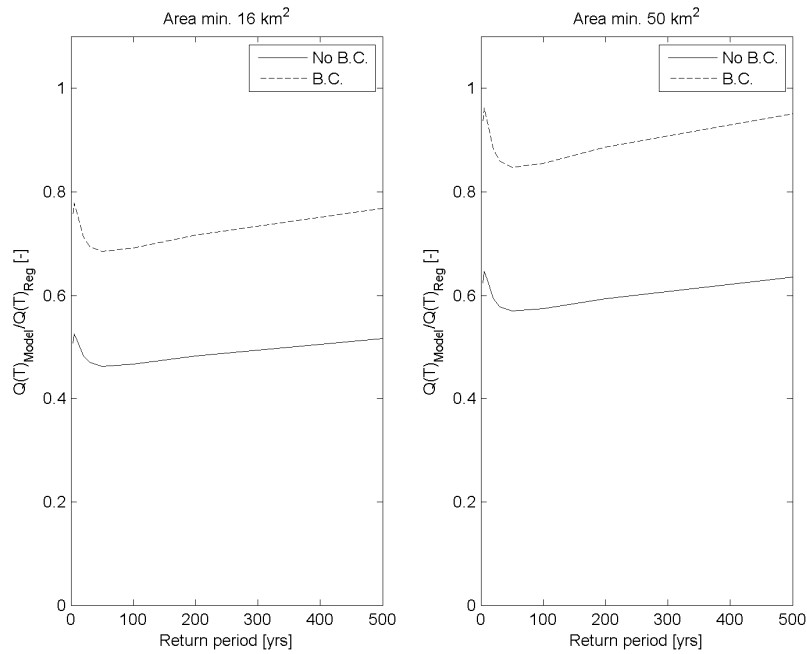

Figure 17. Mean Ratio(T) over the considered domain as a function of T. Continuous line (no
B.C.) is the case without rainfall bias correction, dotted line (B.C.) is the case with rainfall
bias correction. Left panel is the case where points with drainage area lower than 16 km$^2$ are
discarded; right panel is the case where points with drainage area lower than 50 km$^2$ are
discarded



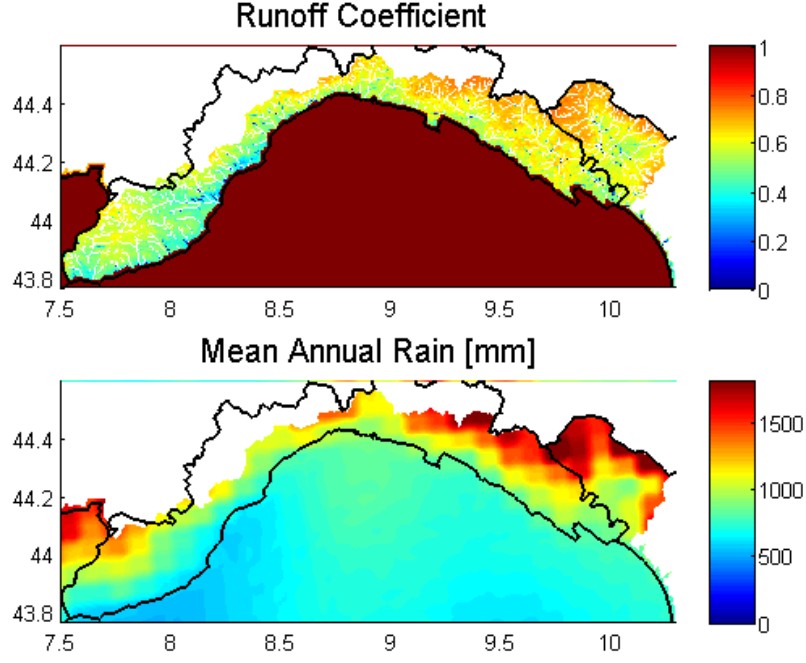

2          Figure 18. Results without rainfall B.C.. Upper panel shows the distributed runoff

3          coefficient while lower panel the mean annual rainfall modeled by EXPRESS-Hydro.





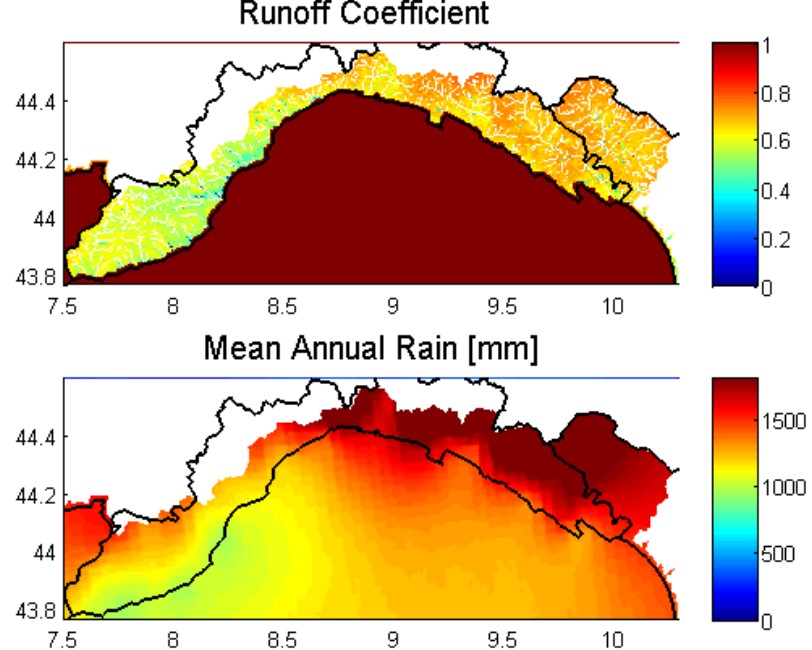

2       Figure 19. Results with rainfall bias correction. Upper panel show the distributed runoff

3       coefficient while lower panel the mean annual rainfall modeled by EXPRESS-Hydro

4       after the rainfall B.C..

