# Peer review of "Analysis of the streamflow extremes and long term water balance"

_Hydrology and Earth System Sciences, 2017_

## Referee Comment (RC1) · Anonymous Referee #1 · 27 Sep 2017

**Manuscript:** Analysis of the streamflow extremes and long term water balance in Liguria Region of Italy using a cloud permitting grid spacing reanalysis dataset, by Silvestro et al.

**General Comments:**

This study has the potential to be a good contribution to the literature; however, it needs to be greatly improved to be accepted. A significant part of the manuscripts is based on the data and analysis provided by Pieri et al., and it is unclear weather some of the analysis showed was performed by Silvestro et al. or not. The structure of the manuscript should be improved; part of the methodology should be moved to introduction and part of the result to methodology (details below). A flowchart describing the methodology would greatly benefit the methodology section. The role of the hydrological model is not sufficiently address, and I think this is a key component that should not be left behind; no matter how good or bad the rainfall input is, if the hydrological model can not properly represent the hydrology, little analysis can be done. For this purpose, more details about model inputs, parameters, processes, uncertainties and performance need to be included. There are too many figure, most can be either removed or combined with others (see below). There are question regarding the usefulness of the regional peakflow analysis, if the purpose of this to be used for prediction in ungauged basins it should be addressed in the manuscript. The English writing style should be revised and improved.

**Specific Comments:**

**Introduction**

- Weak, include more about issues with peakflow estimation and uncertainties, previous studies in the region, hydrological modelling, downscaling and bias correction. Much of the text in other sections should be moved to the introduction (detailed below)
- Page3, line 13-16: I would move it to the methodology section.

**Materials and Method**

**Study Site and Case Study**

Need a better description of the study basins: how many are they and where? Include geomorphological characterization (area, slope, etc), vegetation cover and soil description; are these rivers regulated by dams or do they have extraction, hydrological regime? How is the climatology of this region (precipitation: snow and rain, and temperature?), you should give more details about this. If there are other studies in these basins they should be included and mentioned.

- Page 4, line 11-12: "The response time …" add Reference.
- Figure1: Add to the figure a North Arrow, scale bar, color bar for topography and its source. I would zoom in to the area of interest and add an inset plot showing the regional location of the study site within Europe.
- How many stations from the OMIRL network do you use in the study? What about their quality control, any issues that should be mentioned in this regard? Any snowfall measurements?

- Page 4, line 22: Mention about "historical validated data", what do you mean by that, validated against what exactly? Is there a reference showing the validation?
- Need a better description of the driving data used in the study. Number of stations, a better map showing their location with respect to the basins, available period, gaps in the data if any and variables being recorded (more details).
- Similar for the annual maxima time series, need to include an official identifier number (if available) or name of the hydrometric stations. How reliable are these records? This is particularly important when you are assessing peakflow.

**EXPRESS-Hydro reanalysis**

- Page 5, line 6-20: Rewrite, move to introduction and add references, this is not part of the methodology. Here you should only describe the EXPRESS-Hydro reanalysis.
- Avoid the use of superlative like "very high", instead state the resolution.
- Remove Figure 2, not relevant.
- Add reference for ERA-Interim and WRF.
- Page 6, line 1-17: This is more of a discussion about Pieri et al, I would remove it or synthesize as part of the introduction section. This section should be about the reanalysis' technical features, pros and cons and its suitability for the area of study. What variables are you using from EXPRESS-Hydro? Temporal and spatial resolution? Available period? Etc.

**Bias correction of rainfall fields (B.C.)**

- Replace "Pieri et al (2015) reanalysis" by "EXPRESS-Hydro reanalysis" (everywhere in the text)
- Page 6, line 22-23: no need to include the link for the dataset again.
- Why don't you use the observed dataset as opposed to the corrected EXPRESS-Hydro reanalysis?
- You should not give additional information about EXPRESS-Hydro or observed dataset here; those should be listed in their respective sections.
- Page 8, line 17-18: that's part of the results section
- Did you perform any correction to other weather variables used in the hydrological model, such as air temperature?
- Is there any snowfall in the study basins, was this bias-correction also applied to snowfall? Please clarify
- Is there a particular reason you performed the CDF correction at a monthly scale, how different are the precipitation patterns between months? Maybe with an annual or seasonal correction is enough?
- You need to better argue the use of rainfall station interpolation to bias-correct the reanalysis, spatial interpolation can introduce significant errors in you rainfall estimations, particularly if your monitoring network in sparse, the rainfall regime is very heterogonous or has significant local topographical controls. If any of these apply, then I'm not sure how robust the spatial interpolation is, which can lead to significant problems in the hydrological model performance. You said that regression with other variable, such as elevation, were tested did not show

significant correlation, but is your monitoring network dense enough to argue that? In other words, how is the elevation distribution of your monitoring network? If most of them are at lower elevations then no correlation should be expected.

**Downscaling the precipitation with RainFARM Model**

- Page 9, line 19-22: Could you elaborate more about "runoff formation at small scales". I wonder if there is a study about the runoff mechanisms in this region or if there is something you could include about this. This seems to be rather critical to support your methodology, as this is apparently, the only reason behind applying spatio-temporal downscaling.
- Page 10, line 1-2: Reference

**The hydrological model: Continuum**

- Page 10, line 11-12: Reference
- What's the soil surface temperature used for in the model?
- How is evapotranspiration being simulated by the model? Does the model require air temperature?
- Does the model simulate snowmelt and accumulation?
- Include the units of the parameters.
- This section needs clarification regarding the parameterization approach. There are 6 parameters requiring calibration, did you use the same values for all your basins and landcover? How was the calibration performed, automatically or manually? What was the period used for calibration? Did you calibrate using hourly streamflow measurements? What are the final parameter values for the calibrated and uncalibrated parameters?
- Can you describe the meaning of the REHF index
- Page 12, line 18-19: basins without data for calibration/validation should be removed from the analysis. The empirical nature of the model does not support any parameter transferability; therefore, assuming average values from calibration at other basins should not be used.
- Page 13, line 1-3: clarify, did you run the model for the entire period and used the final conditions from year 2008 as initial conditions for the year 1979?
- What meteorological variables does the model use?

**Results**

**Precipitation analysis**

- Page 13, line 13-20: Apparently, these are not your results should not be in the result section
- Figure 3 to 7: are these your results or from Pieri et al?. If there are not yours you should remove and just reference them. Otherwise you can combine them and only show the difference map for annual and seasonal scales. Note that the mean daily errors are between -3 and 3 mm/d, which for annual rainfall mean an average error of +-1095 mm/yr, which is quite significant and will have a significant impact in your simulations, this should be reconsidered.
- Is Figure 3 using the Bias-corrected EXPRESS-Hydro reanalysis? unclear

- Page 14, line 1-11: These are not your results should not be in the result section
- Unsure whereas what you are showing is from your analysis or Pieri et al. Please clarify.
- Include the corrections applied with the B.C. approach. How large these corrections where?
- Page 14, line 12-17: do this analysis at a basin-scale as that's the relevant scale of the study. I'm not convinced about that EXPRESS-Hydro "reproduces quite well" the observed precipitation. Need further analysis. How are the extreme precipitation events represented by the reanalysis? This is critical to properly represent peakflows. What about temperature?
- Can you evaluate the performance of the spatio-temporal downscaling?, this seems rather critical when assessing peakflow that may be generated from intense hourly rainfall-runoff events.

**Distribution of the annual discharge maxima**

- I think that Fig. 9 to 11 show that model representation of peakflow is somewhat weak, and often the simulations without B.C. show better results (see fig 9 Bisagno La Presa), which is what the Kolmogorov-Smirnov test show as well (only 60% pass the test). This problem can be due to the problems in representing annual rainfall (fig 3-7) and hydrological model!.
- Figure 12 should be replaced by a table.
- Page 17, line 1-23: This is methodology not results, move to the methodology section.
- I think you need to show that the model works well representing ADM at a basin-scale to perform the regional analysis. So far, the simulations at only 9 basins pass the statistical test (out of 15 from fig 12), but then the regional analysis is performed using all of them? Are those basins without streamflow records also included? (they should not). The relatively good agreement from the regional frequency distribution function analysis (fig 13) could be due to a compensatory effect.

**Regional Analysis of the annual discharge maxima**

- Page 17, line 1-15: This should be in the methodology section.
- I suggest expanding about the usefulness of the regional ADM curve, I could see the potential when dealing with prediction in ungauged basins, otherwise I'm not sure what's the purpose of computing this regional curve. The relatively good agreement of this curve against observations showed in figure 13 could be due to some compensatory effect between basins; need to expand on this as well.
- The authors argue that for small scale basins (<50 km2) the simulated ADM curve (Q(T) model) underestimates the regional curve, and attribute this to problems in the reanalysis precipitation quality. I think this analysis is wrong and the fact that the Q(t)model underestimates Q(T)reg (i.e. Ratio(T) <1) only suggests that the regional analysis is not representative of small basins, which could be due to several factors not address in the discussion. (1) If the number of points used to develop the regional curve comes in majority from larger basins, then I wouldn't expect the regional model to represent small scales basins; it is unclear how many large or small scale basins (or grid points) where use to construct the regional curve, this could help the discussion. (2) The role that the hydrological model is playing in representing peakflows is not sufficiently

explored. From the Kolmgorov-Smirnov test, only 9 out of 15 basins past the test (60%), which is not sufficient in my opinion. Problems with model parameterization and process representation can probably explain most of the ADM mismatch for the scenario with bias-corrected rainfall. The representation of peak rainfall events by the reanalysis is unclear, and those are the events that produce peakflows, assuming an entirely rainfall-driven streamflow, which is also unclear from the manuscript. (3) If the EXPRESS-Hydro reanalysis cannot represent the "small-scale" rainfall events, then what was the purpose behind the spatio-temporal downscaling? I think these issues should be better explored and described in the text, particularly regarding uncertainties with hydrological model.

**Water Balance and Runoff coefficient**

- I am not sure about the point of calculating what the authors refer to the "runoff coefficient". I would suggest looking at basin or sub-basin scale runoff ratio (runoff volume/precipitation) as a proxy to the water balance, that way you can avoid storage problems. I don't understand why if all the relevant mass fluxes are being simulated by the model, the authors don't calculate the mass balance directly. I would re-focus this section to a basin-scale mass balance analysis.
- Table 2 shows model streamflow bias, this should be part of a calibration-validation section. No need to show runoff coefficient for the scenario without B.C. as this will clearly be worst than the scenario with B.C.

---

## Referee Comment (RC2) · Anonymous Referee #2 · 16 Oct 2017

**Review of:** Analysis of the streamflow extremes and long term water balance in Liguria Region of Italy using a cloud permitting grid spacing reanalysis dataset

**Authors**: Francesco Silvestro, Antonio Parodi, Lorenzo Campo, Luca Ferraris

**General comments:**

The paper presents a study where hydrologic simulations are performed in Liguria, Italy, through a cascade approach involving the following steps: 1) ERAINT reanalysis are dynamically downscaled using WRF; 2) a bias correction of monthly precipitation is performed through CDF matching; 3) a statistical downscaling model is used to disaggregate precipitation from the coarse resolutions of 4 km, 3 h to the fine resolutions of 1 km, 1 h; 4) the hydrologic model known as Continuum is applied. The modeled discharge is used for three main analyses:

(1) The distributions of the simulated and observed discharge maxima are compared at 15 gauging stations.
(2) A flood frequency regionalization is applied on the simulated discharge values and compared with an existing study based on observations.
(3) The water balance components at annual scale (I guess…) are analyzed.

In my opinion, the study on the discharge maxima can be potentially interesting to assess limitations and potentialities of the proposed approach. However, I have serious concerns on the paper quality and novelty and, at this stage, I recommend its rejection. In the following, I motivate the reasons of my choice.

1) While I am not a native English speaker, I need to point out that the paper is poorly structured and the quality of the English is low. This is a very serious concern that has to be solved even before discussing the technical details. Many sentences are quite long and with poor grammar. In each section, the length of the paragraphs varies too much (from a few lines to an entire page). I suggest the authors to have a native English speaker proofread the paper.

2) The paper structure needs to be improved. The methodology presents results of the hydrologic model calibration; there is a section on the datasets, but new datasets seem to appear in the Results (or maybe they are the same, but named in a different way). There are too many figures that can be merged (see below).

3) The authors need to describe better what is the novelty of their analyses. The idea of applying a cascade-based approach to issue hydrologic predictions has been used in other studies to evaluate climate change impacts (in this case, climate model outputs are used instead of reanalysis) and improve hydrometeorological forecasts (in this case, outputs of Numerical Weather Prediction models are used instead of reanalysis). The Introduction of the paper does not discuss previous applications that follow the same strategy and does not highlight what are the main contributions of this study (in my opinion, it is the analysis of the simulated distribution of flood extremes). In addition, the authors should clearly state which new analyses and simulations have been performed in the paper –I guess these are bias correction, statistical downscaling and hydrologic modeling– or conduced in other studies –the dynamical downscaling of ERAINT.

4) The authors need to present more details on the hydrologic model, including: how the model has

been parameterized, calibrated, and validated with observed data; which soil and vegetation have been used (maybe show a map as well?); which observed hydrometeorological forcings have been adopted and how they have been interpolated in space; and show examples of simulated and observed hydrographs in the calibration and validation periods.

5) A discussion on the performances of the statistical downscaling algorithm is missing.

6) The relatively long discussion of the comparison of the WRF simulations with an alternative rain gauge network to the one used by Pieri et al. (2015) should be shortened. By the way, are the two rain gauge networks completely different?

Comments on the Figures:

- Figure 1: The authors should add legend, scale, and indicate latitude and longitude of the region.

- Figure 2: How is CORDEX related to the experiment of Pieri et al. (2015)?

- Figures 4, 5, 6, and 7 can be merged. I suggest showing also scatter plots and a table with metrics quantifying the performances.

- Figures 15 and 16 can be merged.

---

## Author Comment (AC1) · 24 Oct 2017

Dear reviewer and dear Editor,

In the following we report the answers to the reviewer comments as well assome clarifications about some choices done in the work. We thank the reviewer because we believe that some suggestions can improve the paper. Various comments request to make modifications on manuscript (figures, table, text modifications…),in the answers in some cases we discuss how we propose to modify the text and we give some anticipation about the possible improvements; if editor and reviewer will think that we can proceedwith the review process then we will implement these modifications in the manuscript and submit a new version.

**Introduction**

- Weak, include more about issues with peakflow estimation and uncertainties, previous studies in the region, hydrological modelling, downscaling and bias correction. Much of the text in other sections should be moved to the introduction (detailed below)
- Page3, line 13-16: I would move it to the methodology section.

*We can improve the introduction following the reviewer suggestions, by including more material about previous studies in the region such as:*

- *Boni, G., Ferraris, L., Giannoni, F., Roth, G., & Rudari, R. (2007). Flood probability analysis for un-gauged watersheds by means of a simple distributed hydrologic model. Advances in water resources, 30(10), 2135-2144.*
- *Giannoni, F., Smith, J. A., Zhang, Y., & Roth, G. (2003). Hydrologic modeling of extreme floods using radar rainfall estimates. Advances in Water Resources, 26(2), 195-203.*
- *Silvestro, F., Rebora, N., & Ferraris, L. (2011). Quantitative flood forecasting on small-and medium-sized basins: a probabilistic approach for operational purposes. Journal of Hydrometeorology, 12(6), 1432-1446.*

**Study Site and Case Study**

Need a better description of the study basins: how many are they and where? Include geomorphological characterization (area, slope, etc), vegetation cover and soil description; are these rivers regulated by dams or do they have extraction, hydrological regime? How is the climatology of this region(precipitation: snow and rain, and temperature?), you should give more details about this. If there are other studies in these basins they should be included and mentioned.

*We will expand the description of the study area and its climatology in the new version of the manuscript We provide a general description at region scale because we used it as a continuous territory in modeling, the basins (even considering A>15 km2) are thousands. We could give more information about the study area as requested and add more information in Table 1 for the calibrated basins.*

- Page 4, line 11-12: "The response time …" add Reference.
*We will add it in the new text (Maidment, 1992; Giannoni et al., 2005)*

- Figure1: Add to the figure a North Arrow, scale bar, color bar for topography and its source. I would zoom in to the area of interest and add an inset plot showing the regional location of the study site within Europe.
*We will modify the figure as requested in the new version, follows a new version of the figure (we will modify the caption in the text):*

[Figure]

- How many stations from the OMIRL network do you use in the study? What about their quality control, any issues that should be mentioned in this regard? Any snowfall measurements?
- Page 4, line 22: Mention about "historical validated data", what do you mean by that, validated against what exactly? Is there a reference showing the validation?
- Need a better description of the driving data used in the study. Number of stations, a better map showing their location with respect to the basins, available period, gaps in the data if any and variables being recorded (more details).
- Similar for the annual maxima time series, need to include an official identifier number (if available) or name of the hydrometric stations. How reliable are these records? This is particularly important when you are assessing peakflow.

*Here there is a number of requests regarding data used in the paper. We will collect the available information and improve the description in the text, adding some tables if needed.*

*Regarding "historical validated data": the data are checked by ARPAL personnel in agreement with WMO recommendations to keep out measuring errors and evidently not valid values in order to build a consistent and continuous dataset.*

**EXPRESS-Hydro reanalysis**

- Page 5, line 6-20: Rewrite, move to introduction and add references, this is not part of the methodology. Here you should only describe the EXPRESS-Hydro reanalysis.

*These changes will be done in the new version. We report the new text: "The CORDEX (COordinated Regional climate Downscaling Experiment, Giorgi1 et al. 2009) initiative is aiming at producing regional climate change projections worldwide for input into impact, adaptation and disaster risk reduction studies using regional climate models (RCM) at fine spatio-temporal resolution and forced by different GCMs from the CMIP5 (Coupled Model Intercomparison Project Phase 5, CMIP5) archive. Along these lines Kotlarski et al. (2014) confirmed, with simulations on grid-resolutions up to about 12 km (0.11°), the capability of RCMs to correctly reproduce the main features of the European climate for the period 1989-2008. However they also exhibit relevant modeling errors concerning some metrics, certain regions and seasons: as an example precipitation biases are in the ±40% range while seasonally and regionally averaged temperature biases are generally smaller than 1.5 °C. Building on these findings, Pieri et al. (2015) moved one step further, in the framework of the EXtreme PREcipitation and Hydrological climate Scenario Simulations (EXPRESS-Hydro) project, dynamically downscaling, at 4 km grid spacing and 3 hours time resolution, the historical climate*
* * *
[1] Giorgi, F., Jones, C., & Asrar, G. R. (2009). Addressing climate information needs at the regional level: the CORDEX framework. World Meteorological Organization (WMO) Bulletin, 58(3), 175.

*scenarios generated by the ERA-Interim reanalysis using the state-of-the-art non-hydrostatic Weather Research and Forecasting (WRF) regional climate model."*

- Avoid the use of superlative like "very high", instead state the resolution.
*Agreed, we will replace in the text with "4 km grid spacing"*

- Remove Figure 2, not relevant.
*Agreed, figure will be removed.*

- Add reference for ERA-Interim and WRF.
*Added:*
- *Dee, D. P., Uppala, S. M., Simmons, A. J., Berrisford, P., Poli, P., Kobayashi, S., ... & Bechtold, P. (2011). The ERA-Interim reanalysis: Configuration and performance of the data assimilation system. Quarterly Journal of the royal meteorological society, 137(656), 553-597.*
- *Powers, J. G., Klemp, J. B., Skamarock, W. C., Davis, C. A., Dudhia, J., Gill, D. O., ... & Grell, G. A. (2017). The weather research and forecasting (WRF) model: overview, system efforts, and future directions. Bulletin of the American Meteorological Society, (2017).*

- Page 6, line 1-17: This is more of a discussion about Pieri et al, I would remove it or synthesize as part of the introduction section. This section should be about the reanalysis' technical features, pros and cons and its suitability for the area of study. What variables are you using from EXPRESS-Hydro? Temporal and spatial resolution? Available period? Etc.
*Agreed, this section will be shortened in the new version of the Manuscript, meanwhile it is worth to clarify that Continuum uses from EXPRESS-Hydro the following variables (at 4 km grid spacing and 3 hours time resolution): 2m temperature, 10 m wind, rain depth, downward short wave flux at ground surface, 2m air relative humidity; this is partially done in a previous answer (first comment of this section)*

**Bias correction of rainfall fields (B.C.)**

Replace "Pieri et al (2015) reanalysis" by "EXPRESS-Hydro reanalysis" (everywhere in the text)

*The expression was replaced in the text according to the reviewer request.*

Page 6, line 22-23: no need to include the link for the dataset again.

*The link was removed from the text.*

Why don't you use the observed dataset as opposed to the corrected EXPRESS-Hydro reanalysis?

*The observed data of the monitoring network, as described in Section 2.1, are available with high time frequency, but for a limited number of recent years. On the other side, the historical data from the Atlante Climatico dataset were available for a longer period (about 30 years) but only at daily timestep. A sentence was added in Section 2.1 in order to specify that. Given the fact that the extreme events that occur in the region of study are due to short-duration precipitation events, it was not possible to employ directly these data from the Atlante Climatico dataset, so they were used only for the bias-correction of the EXPRESS-Hydro dataset.*

You should not give additional information about EXPRESS-Hydro or observed dataset here; those should be listed in their respective sections.

*The information about the meteorological datasets was moved in the previous Sections.*

Page 8, line 17-18: that's part of the results section

*This sentence was intended as a feature-by-design of the particular bias-correction procedure that was employed. The sentence was modified in order to be more clear.*

Did you perform any correction to other weather variables used in the hydrological model, such as air temperature?

*Since, as exposed in previous Sections, the extreme flood events in Liguria are caused by short-term, high-intense rainfall events, only precipitation was considered for the bias-correction because it was considered that the contribution to extreme events of other variables, such as the air temperature, was negligible. A sentence was added in the text in order to clarify this.*

Is there any snowfall in the study basins, was this bias-correction also applied to snowfall? Please clarify

*The snowfall in the region of study is present, but not in significant quantity. The contribution of snowfall to the extreme flood events can be considered basically negligible. Furthermore, no snowfall gauges were available in the observed dataset, so the bias-correction was applied on the precipitation.*

Is there a particular reason you performed the CDF correction at a monthly scale, how different are the precipitation patterns between months? Maybe with an annual or seasonal correction is enough?

*As explained in Section 2.2, also with specific reference to Figures numbered as 3-7, strong seasonality is present in both observed (as confirmed by the Atlante Climatico Liguria) and EXPRESS-Hydro datasets. The differences are not negligible also in terms of the single months patterns, thus an annual or seasonal correction would not have been sufficient. A specific sentence about the monthly spatial patterns was added in Section 2.2 in order to specify this.*

You need to better argue the use of rainfall station interpolation to bias-correct the reanalysis, spatial interpolation can introduce significant errors in you rainfall estimations, particularly if your monitoring network in sparse, the rainfall regime is very heterogonous or has significant local topographical controls. If any of these apply, then I'm not sure how robust the spatial interpolation is, which can lead to significant problems in the hydrological model performance. You said that regression with other variable, such as elevation, were tested did not show significant correlation, but is your monitoring network dense enough to argue that? In other words, how is the elevation distribution of your monitoring network? If most of them are at lower elevations then no correlation should be expected.

*The interpolation was employed only at monthly scale, in order to correct the monthly cumulate of the EXPRESS-Hydro dataset. Given the fact that the events of interest are of short duration and spatially concentrated, the error introduced by the interpolation should not affect the structure of such events. The possible distortion introduced by the interpolation can be considered acceptable considering that the raingauge network used for the interpolation itself is sufficiently dense and spatially homogeneous to avoid large areas without sensors. Furthermore, the distribution of the sensors along the different elevations is uniform enough to assure the significance of the correlation analysis. A sentence was added to the text in order to clarify the interpolation issues.*

**Precipitation analysis**
- Page 13, line 13-20: Apparently, these are not your results should not be in the result section
*In this part of the text we compare our analysis with the one done by Pieri et al., so we report a brief summary of their findings. We can better clarify this point and even reduce these comments*

- Figure 3 to 7: are these your results or from Pieri et al?. If there are not yours you should remove and just reference them. Otherwise you can combine them and only show the difference map for annual and seasonal scales. Note that the mean daily errors are between -3 and 3 mm/d, which for annual rainfall mean an average error of +-1095 mm/yr, which is quite significant and will have a significant impact in your simulations, this should be reconsidered.

*These figures are part of our work, we will better clarify this point in the text. As shown in some part of the region errors are quite large. For this reason we applied a rainfall bias correction. Still it is worth to mention that errors are definitely lower than those corresponding to coarser grid spacing dynamically downscaled (e.g. 12 km) reanalysis.*

- Is Figure 3 using the Bias-corrected EXPRESS-Hydro reanalysis? unclear
*Without B.C. We will better clarify this in the caption.*

Page 14, line 12-17: do this analysis at a basin-scale as that's the relevant scale of the study. I'm not convinced about that EXPRESS-Hydro "reproduces quite well" the observed precipitation. Need further analysis. How are the extreme precipitation events represented by the reanalysis? This is critical to properly represent peakflows. What about temperature?

*We propose to editor and reviewer to make the analysis on some basins along the study area just to give an idea of the possible variability of the results when going to basin scale.*
*In fact doing the presented analysis (now done at regional scale) at basin scale it's not feasible since we work with a distributed approach and in such a sense the concept of basin becomes unuseful (each point of the domain has an upstream basin).*
*Concerning the other questions, Pieri et al. (2015) assessed the EXPRESS-HYDRO results capability to correctly reproduce the statistics of intense precipitation events both over Europe CORDEX domain at large and over the Great Alpine Region (GAR). The following figure shows the map of the number of days with rainfall rate larger than 10 mm for E-OBS and for EXPRESS-HYDRO runs. According to E-OBS, frequent heavy precipitation days are essentially localized over the Alps, western Norway, Scotland, and Portugal. This spatial distribution is well captured by EXPRESS-HYDRO runs.*

[Figure]

**Figure 1: Number of days with heavy precipitation (>10 mm) for the period 1979–98 (modified from Pieri et al. 2015).**

*Pieri et al. (2015) investigated the distribution of the probabilities of exceedance of precipitation thresholds for the daily rainfall rate on the GAR domain, for all seasons, for the 12 and 4 km grid spacing model results and for EURO4M-APGD observational dataset. The high-resolution run at its original 4 km grid spacing agrees well with observations in most seasons but underestimates the probabilities of intense precipitation (at about > 100mm/day) when aggregated at 12 km, except in winter when the aggregated high-resolution run is close to the observations.*

[Figure]

FIG. 7. Exceedance probability for 1979–98 over the GAR, for different WRF simulations and for EURO4M-APGD, for seasons (a) DJF, (b) March–May (MAM), (c) JJA, and (d) September–November (SON). The exceedance probability is computed as 1 minus the precipitation cdf.

Can you evaluate the performance of the spatio-temporal downscaling?, this seems rather critical when assessing peakflow that may be generated from intense hourly rainfall-runoff events.

*The testing and application of RainFARM in the study area was done in many works, some already cited in the text ( Rebora et al. 2006; Silvestro et al. 2012). As stated by reviewer this is a potentially critical point and we discussed the effects on the very small basins in the text and in the following answers to the comments. We will also improve comments on text.*

*To assess the impact of downscaling we made the hydrological simulation with BC rainfall but **without** applying the downscaling, we estimated the ratio of Qmean (mean of 30 years ADM) without and with Downscaling for each pixel.*

*Results are plot versus drainage area (see figure below). It is quite evident the impact of downscaling, the impact generally increases when drainage area decreases. This graph also help to answer to one of the comment below regarding the underestimation of quantiles for very small basins: the downscaling is crucial to face this issue but probably the adopted configuration is not sufficient when drainage area drops below certain values.*

[Figure]

**The hydrological model: Continuum**

- Page 10, line 11-12: Reference

*We have not scientific reference regarding this point apart for the experiment in Silvestro et al. 2013, but we did some operational implementations in specific contracts. If editor and reviewer believe that we have to remove this sentence we can do it.*

- What's the soil surface temperature used for in the model?

*As mentioned in the text (page 10, line 7) LST is an output of the model*

- How is evapotranspiration being simulated by the model? Does the model require air temperature?

*Yes, in the revised text we added the following sentence:" Continuum needs as input the following variables: rainfall, air temperature, short-wave solar radiation, wind velocity, air relative humidity."*

- Does the model simulate snowmelt and accumulation?

*The model simulates snow dynamics, we added a sentence in paragraph 2.5 : "Snow melting and accumulation is simulated with simple equations forced with air temperature and solar radiation(Maidment, 1992) as described in Silvestro et al. (2015)."*

- Include the units of the parameters.

*Done in the new version*

- This section needs clarification regarding the parameterization approach. There are 6 parameters requiring calibration, did you use the same values for all your basins and landcover? How was the calibration performed, automatically or manually? What was the period used for calibration? Did you calibrate using hourly streamflow measurements? What are the final parameter values for the calibrated and uncalibrated parameters?

*We added some more information and a new table (Table 2) with final parameter values as requested. "NS and REHF were combined in a multi-objective function to carry out the calibration using the approach proposed in Madsen (2000). Calibration was done analyzing the parameter space using a brute force*

*approach on 2011-2013 period in order to find the parameter set that optimize the multi-objective function. Curve Number map is derived by the CORINE Land cover (http://www.sinanet.isprambiente.it/it/progetti/corine-land-cover-1).The final setting is similar to the one described in Davolio et al. (2017)."*

- Can you describe the meaning of the REHF index
*We added the following sentence in section 2.5: "While NS aims to assess the general reproduction of streamflow, REHF score has the aim of giving high weight to high flows leading the calibration to better reproduce the flood events."*

- Page 12, line 18-19: basins without data for calibration/validation should be removed from the analysis. The empirical nature of the model does not support any parameter transferability; therefore, assuming average values from calibration at other basins should not be used.
*We know that this is not the optimal condition of work, it is clear that it would be better to have a very large number of calibrated basin, but this is also quite common in real application.*
*We only partially gree with this comment for at least two reasons:*
i)      *the Continuum hydrological model (and its predecessor DRiFt, that was designed only for simulating floods at event scale) demonstrated to well work in the study environment (Giannoni et al., 2000, 2005; Gabellani et al., 2008; Silvestro et al 2013, 2015, Cenci et al 2016...) giving reasonable and reliable results also for uncalibrated sections when used for flood forecast aims (Regione Marche, 2016). This is valid when the basins have similar characteristics especially regarding the surface response to intense rainfall events and main genesis of rainfall-runoff process, as a consequence parameters have often similar values moving from a catchment to another. We agree with reviewer that if we would use Ligurian average parameters in other environments, with completely different characteristics (example flat basins,) results could be affected.*
ii)     *Even the benchmark regional analysis (Boni et al., 2007) is done with the model DRiFt which was not possible to calibrate in each place of the target region. The hydro model is used indeed to have flow time series in un-gauged basins*
*We can stress the problem of calibration in the text but we believe it would be interesting to keep the analysis over the entire region.*

- Page 13, line 1-3: clarify, did you run the model for the entire period and used the final conditions from year 2008 as initial conditions for the year 1979?
*No, we modified the sentence in order to better clarify:" In order to have reduced warm up impacts on 1979 simulation a first run was done starting from a predefined initial condition and the state variables simulated on the 31st of December of every year (from 1979 to 2008) were used averaged to estimate a reasonable initial condition for 1st January 1979 to be used in the final simulation.*

- What meteorological variables does the model use?

*We added the following sentence in section 2.5: "Continuum needs as input the following variables: rainfall, air temperature, short-wave solar radiation, wind velocity, air relative humidity."*

**Distribution of the annual discharge maxima**
- I think that Fig. 9 to 11 show that model representation of peakflow is somewhat weak, and often the simulations without B.C. show better results (see fig 9 Bisagno La Presa), which is what the Kolmogorov-Smirnov test show as well (only 60% pass the test). This problem can be due to the problems in representing annual rainfall (fig 3-7) and hydrological model!.
*We could add some comments regarding the results; yes, as already mentioned in the text, the fact that results are not optimal in some sections are probably due to a combination of different causes: i) sometimes and somewhere the rainfall reanalysis is probably poorly representative of real rainfall and B.C. does not*

*correct it enough ii) hydrological model setting could be not optimal in some basins iii) observed peak simulated and peaks are often referred to different time periods, this fact together to the hydro-climatic regime typical of the study region (flash flood regime with high variability of ADM) could have a certain impact on final results.*

*The study is not aiming to show that the applied chain reproduces perfectly observations and benchmark (Boni et al., 2007 for regiona analysis) but the overarching goal is to combine, for the first time of the best of authors knowledge in a complex orography area such as Liguria, a cloud-permitting grid spacing reanalysis together with a state of the art hydrological modelto understand what is possible to do with such kind of chain, and thus offering a methodological framework for future similar analysis.*

*In any case we still believe that the results are interesting and not so weak even in respect to the hydrometerological characteristics of the study region (frequent occurrence of flash floods).*

- Figure 12 should be replaced by a table.
*Agreed, it will be done in the new version of the manuscript*

- Page 17, line 1-23: This is methodology not results, move to the methodology section.
*Agreed, it will be done in the new version of the manuscript*

- I think you need to show that the model works well representing ADM at a basin-scale to perform the regional analysis. So far, the simulations at only 9 basins pass the statistical test (out of 15 from fig 12), but then the regional analysis is performed using all of them? Are those basins without streamflow records also included? (they should not). The relatively good agreement from the regional frequency distribution function analysis (fig 13) could be due to a compensatory effect.

*We agree only partially with this approach (in brief: not considering streamflow series in un-calibrated sections) and we tried to explain our point of view (which is also based on previous publications). It is, in our opinion and based on our scientific experience, indeed interesting to look at the results in a distributed perspective using modeled time series to reproduce streamflow where no observations are available (being conscious of the possible uncertainty and errors of simulated time series). This is an approach similar to the one done in a previous scientific papers, in particular the one that describe our benchmark: Boni et. al (2007). The benefit of using a large number of time series is often larger than the uncertainty introduced by not optimal parameter set, especially when ADM are normalized and used to build a regional curve.*

*If the reviewer and editor agree we propose to show also the regional curve obtained using only the 9 basins which passes the test. This comparison could help to show the possible differences. The following figure is generated using time series from the 9 basins which pass te KS test, the fit is good as the one shown in the paper.*

[Figure]

Moreover it is worth to highlight that in many cases (Figures 9 to 12) simulated ADM distributions have similar shapes to observed ones and suffer of a sort of bias (probably driven by errors in rainfall or hydro-model), in other cases the simulated ADM distribution is only partially out of the confidence interval, but hydro-climatology seems to be discretely reproduced. This is a probable motivation because regional curve is quite well represented. For example, in the La Presa case the evident Bias is presumably lead by one of the effects presented in section 3.3, we report the text: "……it is possible that EXPRESS-Hydro well reproduces the events at small time and spatial scales (3-6 hours, 10-100 km2) in that part of the region but generally underestimates monthly cumulates, in this case B.C. could lead to streamflow overestimation….". When considering the Normalized values they contribute to good results in building regional curve.

**Regional Analysis of the annual discharge maxima**

- Page 17, line 1-15: This should be in the methodology section.

*Agreed, we will move this part on a new paragraph (2.7) in section 2, in the new version of the manuscript.*

- I suggest expanding about the usefulness of the regional ADM curve, I could see the potential when dealing with prediction in ungauged basins, otherwise I'm not sure what's the purpose of computing this regional curve. The relatively good agreement of this curve against observations showed in figure 13 could be due to some compensatory effect between basins; need to expand on this as well.

*We will insert some more comments about the usefulness of the regional curve in the new version of the paper. The relatively good agreement could be also due the aforementioned motivation: even if in some*

*basins the fit is not optimal the simulated ADM reproduce discretely the typical hydrological regime of the study region.*

- The authors argue that for small scale basins (<50 km2) the simulated ADM curve (Q(T) model)underestimates the regional curve, and attribute this to problems in the reanalysis precipitation quality. I think this analysis is wrong and the fact that the Q(t)model underestimates Q(T)reg (i.e.Ratio(T) <1) only suggests that the regional analysis is not representative of small basins, which could be due to several factors not address in the discussion. (1) If the number of points used to develop the regional curve comes in majority from larger basins, then I wouldn't expect the regional model to represent small scales basins; it is unclear how many large or small scale basins (or grid points) where use to construct the regional curve, this could help the discussion.(2) The role that the hydrological model is playing in representing peakflows is not sufficiently explored. From the Kolmgorov-Smirnov test, only 9 out of 15 basins past the test (60%), which is not sufficient in my opinion. Problems with model parameterization and process representation can probably explain most of the ADM mismatch for the scenario with bias-corrected rainfall. The representation of peak rainfall events by the reanalysis is unclear, and those are the eventsthat produce peakflows, assuming an entirely rainfall-driven streamflow, which is also unclear from the manuscript. (3) If the EXPRESS-Hydro reanalysis cannot represent the "small-scale" rainfall events, then what was the purpose behind the spatio-temporal downscaling? I think these issues should be better explored and described in the text, particularly regarding uncertainties with hydrological model.

*We will add some considerations based on the comments of the reviewer: they are potentially good reasons and clear explanations for underestimation in very small basins, thus we will improve the discussion of the results in this respect.*

*As a general comment we would like to highlight again that we are exploring the potentialities of the presented modeling chain to reproduce ADM in an environment made by small basins, in complex topography areas, especially testing the usage of a high resolution (cloud permitting) reanalysis. To the best of our knowledge this is one of the first works of this kind done in the considered study area and we have not the claim to solve all the possible issues and problems in a unique work, or to define a definitive methodology that can be applied in any case. On the other side the fact that in some cases the system cannot reproduce in optimal way observations (..or better we should say their estimation..) cannot be considered a bad result or weak point, simply the analysis shows that in some cases the system does not work optimally; in many others the results are good, 9 out of 15 basins are working well also accounting that we are working only with models and only observations on very coarse time scale are used in the system (for Bias correction).*

*We will improve discussion and motivation but we do not believe that our analysis is totally unuseful. The model was calibrated and we have a consolidated experience in applying it in the studied environment (we already cited various works), so it appears unlikely that a possible not perfect parameterization leads to such a general underestimation on small basins (more probable a more uniform distribution between over and underestimation); anyway the model spatial resolution could play a role since the representativeness of morphology degrades for small drainage areas with general smoothing effect that affects results; we put a threshold for the analysis (basins with Area<15 km$^2$ are neglected) but the degradation effect is clearly continuous from larger to smaller drainage areas. But there is surely another issue to account for: the used time resolution (1 hour after downscaling) could be not sufficient in some cases for these basins, various works (example Silvestro et al., 2016; Rebora et al., 2013) and the experience prove that rainfall time*

*pattern on scales lower than 1 hour have significant impact on high flows for such small basins. We already commented this fact in the text (pgg 18 lines 16-18 and also in section 4: Discussion and conclusions). The combination of these effects could lead to a flow underestimation (that reflects on the index flow which is estimated as average of the 30 year series of ADM) when drainage area decreases. This effect is compensated in the centre of the region where B.C. seems to lead to a general overestimation (see also comments in the text) of quantiles. Figure 7 also seems to support this motivation.*

*The regional curve is built using simulated time series for cells with upstream area>15 km² (pgg 13 lines 6-10), so it should not be affected by preponderance of large basins.*

*As we discussed in the text, probably the results demonstrate that very small basins should be treated with a different setting of the system and a dedicated work (even, as mentioned in the text, trying to exploiting more deeply the downscaling potentialities) , we would expand the discussion adding all these considerations. (See also response and figure to comment regarding the downscaling)*

**Water Balance and Runoff coefficient**

I am not sure about the point of calculating what the authors refer to the "runoff coefficient". Iwould suggest looking at basin or sub-basin scale runoff ratio (runoff volume/precipitation) as aproxy to the water balance, that way you can avoid storage problems. I don't understand why ifall the relevant mass fluxes are being simulated by the model, the authors don't calculate themass balance directly. I would re-focus this section to a basin-scale mass balance analysis.

*The "runoff coefficient" is exactly the runoff ratio and we can change the name in the text as suggested. We did the analysis of runoff ratio because, as mentioned in the text, we have a term of comparison from Hydrologic Annual Survey. For the Analysis at pixel scale we used the first formulation showed in section 3.4 because rain and evt are the standard output and input available for common hydrological models (also for Continuum), while other components (input and output of each cell) are not saved as output. It's clear that the mass balance at single cell scale is closed. Long term runoff ratio can be estimated in both ways (function of P and Q, or function of P and Evt). We will better clarify these points.*

Table 2 shows model streamflow bias, this should be part of a calibration-validation section. No need to show runoff coefficient for the scenario without B.C. as this will clearly be worst than the scenario with B.C.

*We will try to better clarify the sense of these results in the text. Runoff coefficient (..or  runoff ratio) are obtained using Express HYDRO as input, so they cannot be considered part of calibration (which is done using observations on a period where they are continuously available) but are part of the hydrological reanalysis. We still believe interesting to keep both results (with and without B.C.) because they show the impact of rainfall B.C. but also of the other variables (there are already comments about this on page 22 lines 11-17); anyway if the editor and reviewer believe that results without B.C. are unuseful we can remove them*

**References**

Cenci L., Laiolo P., Gabellani S., Campo L., Silvestro F., Delogu F., Boni G., and Rudari R., "Assimilation of H-SAF Soil Moisture Products for Flash Flood Early Warning Systems. Case Study: Mediterranean Catchments", IEEE Journal of Selected Topics in Applied Earth Observations and Remote Sensing, vol. 9, no. 12, pp. 5634-5646, Dec. 2016. DOI: 10.1109/JSTARS.2016.2598475

Davolio, S., F. Silvestro, and T. Gastaldo, 0: Impact of rainfall assimilation on high-resolution hydro-meteorological forecasts over Liguria (Italy). J. Hydrometeor., 18, 2659-2680,https://doi.org/10.1175/JHM-D-17-0073.1

Giannoni, F., Roth., G., and Rudari, R.: A Semi – Distributed Rainfall – Runoff Model Based on a Geomorphologic Approach. Physics and Chemistry of the Earth, 25/7-8, 665-671, 2000.

Giannoni, F., Roth, G. Rudari, R.: A procedure for drainage network identification from geomorphology and its application to the prediction of the hydrologic response, Advances in Water Resources, 28(6), 567-581, 2005, doi:10.1016/j.advwatres.2004.11.013.

Regione Marche. -Regionalizzazione delle portate massime annuali al colmo di piena per la stima dei tempi di ritorno delle grandezze idrologiche. http://www.regione.marche.it/Regione-Utile/Protezione-Civile/Progetti-e-Pubblicazioni/Studi-Meteo-Idro#Studi-Idrologici-e-Idraulici. Last access date:19/10/2017

Rebora, N., Molini, L., Casella, E., Comellas, A., Fiori, E., Pignone, F., Siccardi, F., Silvestro, F., Tanelli, S., Parodi, A.: Extreme Rainfall in the Mediterranean: What Can We Learn from Observations? J. Hydrometeorol., 14, 906-922, 2013.

Silvestro, F., Rebora, N., Giannoni, F., Cavallo, A., Ferraris, L., (2016) The flash flood of the Bisagno Creek on 9th October 2014: an "unfortunate" combination of spatial and temporal scales. Journal of Hydrology, 541, Part A, Pages 50–62, doi:10.1016/j.jhydrol.2015.08.004

---

## Author Comment (AC2) · 24 Oct 2017

Dear reviewer dear Editor,

In the following we report the answers to the comments and the clarifications about some choices done in the work. Various comments request to make modifications on manuscript (figures, tables, text modifications…), in the answers we discuss how we propose to modify the text and we give some anticipations about them. We think the reviewer suggestions are in various cases useful to improve the work, anyway despite the reviewer seems to think that the analysis is interesting in the complex, he issued a **rejection** final evaluation; as a consequence we do not know if we have the chance to submit a new version. We ask to editor and reviewer if they think that we can go on with the review process.

In my opinion, the study on the discharge maxima can be potentially interesting to assess limitations and potentialities of the proposed approach. However, I have serious concerns on the paper quality and novelty and, at this stage, I recommend its rejection. In the following, I motivate the reasons of my choice.

1) While I am not a native English speaker, I need to point out that the paper is poorly structured and the quality of the English is low. This is a very serious concern that has to be solved even before discussing the technical details. Many sentences are quite long and with poor grammar. In each section, the length of the paragraphs varies too much (from a few lines to an entire page). I suggest the authors to have a native English speaker proofread the paper.

*We can review the general quality of language and writing, moreover since we are not native English speakers we often ask to native English speaker to improve the language but this is always occurred during an advanced phase of the review.*

*Since all the authors has a certain experience in publishing papers we were surprised to realize that this work is such poorly readable, anyway we will follow the reviewer suggestions to improve the text.*

2) The paper structure needs to be improved. The methodology presents results of the hydrologic model calibration; there is a section on the datasets, but new datasets seem to appear in the Results (or maybe they are the same, but named in a different way). There are too many figures that can be merged (see below).

*Agreed, this can be done since this is a request similar to those made by reviewer 1. Our intention is to improve the description and naming of data and give more detail about hydrological model. Some parts of the paper can be restructured by moving some sections in different parts of the manuscript. Some figures can be merged.*

3) The authors need to describe better what is the novelty of their analyses. The idea of applying a cascade-based approach to issue hydrologic predictions has been used in other studies to evaluate climate change impacts (in this case, climate model outputs are used instead of reanalysis) and improve hydrometeorological forecasts (in this case, outputs of Numerical Weather Prediction models are used instead of reanalysis). The Introduction of the paper does not discuss previous applications that follow the same strategy and does not highlight what are the main contributions of this study (in my opinion, it is the analysis of the simulated distribution of flood extremes). In addition, the authors should clearly state which new analyses and simulations have been performed in the paper –I guess these are bias correction, statistical downscaling and hydrologic modeling– or conduced in other studies –the dynamical downscaling of ERAINT.

*It is true that the similar modeling cascade were already applied, but to the best of our knowledge this is one of the first works of this kind done in the considered study area in which the environment is constituted by small basins with flash flood hydrological regime. Moreover the usage of such high resolution reanalysis is quite new. We can highlight this point in the text as suggested, even if the introduction already mentions this fact: "….On one side the distributed nature of the state and output variables allowed to investigate the possibility of using this kind of modelling chain for extreme streamflow statistical analysis (e.g. distribution of annual discharge maxima) and long term water balance (e.g. long term runoff coefficient) in a fully distributed perspective. On the other side the high spatial grid spacing and time resolution of forcings,*

*together with the use of a rainfall downscaling model, allowed to explore the use of such high resolution reanalysis in regions characterized by the presence of small hydrological watersheds in areas characterized by very complex topography..."*
*We can also improve the discussion of previous works that use similar strategies, but this is already present. In lines 3 to 12 of page 3 we cited various works that involve usage of reanalysis for hydrological purposes.*

4) The authors need to present more details on the hydrologic model, including: how the model has been parameterized, calibrated, and validated with observed data; which soil and vegetation have been used (maybe show a map as well?); which observed hydrometeorological forcings have been adopted and how they have been interpolated in space; and show examples of simulated and observed hydrographs in the calibration and validation periods.

*Agreed, this is also a request of the reviewer 1. Moreover we have a recent published paper to be cited where we used a similar setting of the model on the same study area for different purpose (Davolio et al., 2017). We did not devote too much space regarding the calibration of the model because it was partially faced in other works and it is not the core of the paper but it is a "functional" activity for the other analysis. Some information are already present in the text, for example regarding interpolation : "….ground stations measurements were interpolated on a regular grid of 1 km resolution by using Kriging method and used as input to the model...."*
*To summarize the performance of the model we inserted a table with the skill scores value. If the editor and the reviewer consider it necessary some graphs of model versus observations we can do that, but, in our opinion, this appears a little bit out of the scope of the paper and in contrast with the need to reduce figures number.*

5) A discussion on the performances of the statistical downscaling algorithm is missing.
*The testing and application of RainFARM in the study area for forecast purposes was done in many works, some already cited in the text ( Rebora et al. 2006; Silvestro et al. 2012). In this work it is used, as mentioned in the text, to generate a possible downscaled rainfall scenario from 4km-3hours to 1km-1hour resolution. Effectively we did not test the benefit of using or not the downscaling.*
*To assess the impact of downscaling we made the hydrological simulation with BC rainfall but **without** applying the downscaling, we estimated the ratio of Qmean (mean of 30 years ADM) without and with Downscaling for each pixel.*
*Results are plot versus drainage area (see figure below). It is quite evident the impact of downscaling, the impact generally increases when drainage area decreases. The downscaling seems to be important even in this application especially to deal with small basins.*

[Figure]

6) The relatively long discussion of the comparison of the WRF simulations with an alternative rain gauge network to the one used by Pieri et al. (2015) should be shortened. By the way, are the two rain gauge networks completely different?
*We can better clarify this point in the text. The raingauge data set used in Pieri et al. (2015) is a subset of the one used in our work.*

Figures.
*We can reduce and improve the figures following the reviewer suggestions.*

**References**

Cenci L., Laiolo P., Gabellani S., Campo L., Silvestro F., Delogu F., Boni G., and Rudari R., "Assimilation of H-SAF Soil Moisture Products for Flash Flood Early Warning Systems. Case Study: Mediterranean Catchments", IEEE Journal of Selected Topics in Applied Earth Observations and Remote Sensing, vol. 9, no. 12, pp. 5634-5646, Dec. 2016. DOI: 10.1109/JSTARS.2016.2598475

Davolio, S., F. Silvestro, and T. Gastaldo,: Impact of rainfall assimilation on high-resolution hydro-meteorological forecasts over Liguria (Italy). J. Hydrometeor., 18, 2659-2680, 2017. https://doi.org/10.1175/JHM-D-17-0073.1

Giannoni, F., Roth., G., and Rudari, R.: A Semi – Distributed Rainfall – Runoff Model Based on a Geomorphologic Approach. Physics and Chemistry of the Earth, 25/7-8, 665-671, 2000.

Giannoni, F., Roth, G. Rudari, R.: A procedure for drainage network identification from geomorphology and its application to the prediction of the hydrologic response, Advances in Water Resources, 28(6), 567-581, 2005, doi:10.1016/j.advwatres.2004.11.013.

Rebora, N., Molini, L., Casella, E., Comellas, A., Fiori, E., Pignone, F., Siccardi, F., Silvestro, F., Tanelli, S., Parodi, A.:  Extreme Rainfall in the Mediterranean: What Can We Learn from Observations?  J. Hydrometeorol., 14, 906-922, 2013.

Silvestro, F., Rebora, N., Giannoni, F., Cavallo, A., Ferraris, L., (2016) The flash flood of the Bisagno Creek on 9th October 2014: an "unfortunate" combination of spatial and temporal scales. Journal of Hydrology, 541, Part A, Pages 50–62, doi:10.1016/j.jhydrol.2015.08.004

---

## Referee Report (RR1)

The reviewed manuscript corresponds to the authors response to the mayor revisions of the article: "Analysis of the streamflow extremes and long term water balance in Liguria Region of Italy using a cloud permitting grid spacing reanalysis dataset" by Silvestro et al.

**General comments**

- The authors present an approach (modelling chain) to use a long-term and high spatial resolution (4km) regional climate model with further bias correction to force a hydrological model over a large region in Italy. Calibration of the hydrological model was performed when possible and then parameters were transferred to ungauged basins. On gauged basins the model seems to represent relatively well streamflow; however, I see a significant limitation in the approach used to transfer these parameters to ungauged basins, as these are purely empirical, and the average of the calibrated parameters were used in ungauged. A GEV distribution was fitted to the simulated Annual Discharge Maxima (ADM) at all gauged sites and compared with observed streamflow, for which I see a fair agreement as fitted GEV 95% confidence interval sometimes does not cover observations (3 out of 6 in my opinion). I think the authors underestimate the role of the hydrological model in the overall analysis and discussion through the manuscript; more emphasis should be given to it, as this is largely driving the simulated hydrological response of these basins. The Ratio(t) is defined to assess the performance of a regionalization approach in characterizing streamflow for several return periods (T). Honestly I have somewhat a hard time understanding the usefulness of this regionalization approach, as it does not seems to work very well. I encourage the authors to demonstrate that the Ratio(T) is being well represented, as I see problems in small and large scale basins (Figure 11). The analysis of the effect of downscaling precipitation on the streamflow should be revised (see specific comments). The final analysis of the mass balance components (runoff ratio) seems very useful and maybe it should be more deeply explored and the main focus of the manuscript.
- I acknowledge that the authors responded to most of the comments from the previous revision; however, there are still significant issues with the manuscript in my opinion.
- For latter revisions I would encourage adding more details to the author's response letter, specifically where the changes are located in the manuscript (page and line), that way it would be much easier for us to go through the revisions.
- I believe that the English of the manuscript needs to be further improved.

**Specific Comments:**

**1 Introduction:**

- Page 4, Line 22-23: I agree in the use of high-resolution RCM to reproduce small-scale rainfall events, but I think you need a reference to this or even better if you can show that this is the case with the original dataset in your study region.

**2 Material and Methods:**

**Study area and case study:**

- Figure 1: To help the description of the figure, use (a), (b) and (c) to refer to each of the plot. And explain each of them as an inset of the others. I'm not sure about the utility of including the Curve Number, if you want to keep this you need to reference the source of this information. Also for the legend use "Curve Number" instead of C.N., try to avoid acronyms, same for DEM, etc.
- Page 5, line 12: refer to table 1.
- Page 6, line 5: What about the raingage distribution vs elevation, is orographic precipitation well represented by these stations? Discuss.
- Page 6, line 5-7: be more specific, how low?
- Page 6, line 7-8: what data was used for calibration/validation?, I assume streamflow discharge, but it is unclear.

**Downscaling the rainfall**

- Pag 10, line 10: Define Lr and tr.
- Pag 10, line 10-12: add reference.

**The hydrological model**

- This section needs better organization, use subsections if needed to explain: model inputs, calibration/validation and model structure (parameters, spatial resolution, etc.)
- How is evapotranspiration being calculated?
- pag 13, line 15-17: Calibration procedure is not very clear. What do you mean by 11 sections, what's a "section" (basin?). What ground stations measurements were interpolated? I assume you calibrate and validate the model using the downscaled reanalysis data, please clarify.
- At what temporal scale did you calculated NS and REHF, hourly or daily?
- What range did you use for calibration and why?, explain. How did you come up with the 500 mm and 1 (Vxmax and Rf) for all the basins, wild guess?
- Pag. 14, line 16-17: This point is crucial in the manuscript. I disagree that taking average parameters values from calibration into ungauged basins is a satisfactory approach, especially as this are highly empirical values, and this is critical to analyze the results. I would encourage the authors to demonstrate that the impact of taking average parameters in ungauged basins is minimal. For this purpose you can use the average parameter values in the model of a gauged basin and demonstrate that these parameters have little impact on the streamflow performance. I think a significant assumption is being made here, which is not easy to support given the empirical basis of the model.
- For the regional analysis it is necessary to know how much of your study region is covered by gauged and ungauged basins, this can help discussing the impact of the assumptions in model's parameters.
- Table 5: need to include the periods available for discharge, and ADM.

**3 Results**

- Figure 2 and 3: I would recommend showing the difference in precipitation as a percentage with respect to the observed precipitation, or as (mm), so it is easier to compare with the other maps which are in mm.
- Pag 17, line 17: not sure what you mean by "are largely confirmed"?. Please change the units of the difference map so the comparison is easier. And do the same when you analyze this difference.
- Figure 4: Include coefficient of determination of correlation and mean bias.
- Figure 5 and 6: Values presented in this figures are good for the analysis; however, I think you already have enough figures regarding the performance of EXPRESS-Hydro (Fig 2 to 6) I would try to merge or remove some, maybe by adding more information to Table 3 you can remove one or two plots without losing much information for the analysis.
- As ADM are a key aspect of your analysis, I would also show the performance of the bias-corrected precipitation time-series in representing peak precipitation, you could look at this by comparing the probability distribution functions of simulated and observed precipitation. I think this is very important as your peak flows are driven by the peak precipitation events, not by the precipitation volume. This will help you in the discussion of the ADM model's performance.
- Pag 18, line 21-24: why did you chose these 4 basins and nor others?
- You should try to merge figure 7, 8 and 9.
- Why did you choose these 4 basins to contrast the analysis?
- Explain what is that you are testing with Kolmorogov-smirnov test, and this should be in your methods too. I know you are talking about ADM distribution but it should be clear from in your manuscript.

**Section 3.3**

- Figure 10: are you using data from every grid point in your model to construct this curve or selected basin outlet?, this need to be clarified in the text. Avoid shortening the words in the figure; you have enough room in the plot, change 'Calib' to calibration, 'simul' to simulations, etc. This applies to all plots/tables.
- Can you comment or interpret the step-changes in the observed growth curve of figure 10 in terms of the dominating hydrological processes in the region, and why you don't see that in the simulated values?
- It seems to me that the relatively better performance in the regional growth curve, both calibrated and total area, could be the results of errors compensation as figure 7 to 9 show that observed distribution of ADM lies outside of the confidence boundary in some cases (Bisagno, Magra, Argentina; 3 of 6). This needs more discussion in the manuscript. If this approach is meant to be used in ungauged basins, the model needs to prove that it works in the calibrated basins. This can be a major problem in the proposed modelling chain. If the authors can prove that the model in calibrated basins can represent ADM, this will help a lot in supporting the proposed modelling chain.

- Figure 11 shows that small basins are underestimated (as discussed in the manuscript), and it also shows an overestimation for the Ratio(T) for large basins, how can the authors explain this?. Also I'm not convinced that the B.C. Ratio(T) shows an improvement in Figure(11), as small basins that used to have a good performance (near 1) now they are overestimating (Ratio>1). Overall I'm doubtful about the usefulness of this Ratio(T) as it does not seem to produce good results. I think the authors need to show that this index, which tries to show that the regional curve is suitable for all basins, works.
- Figure 12 has little discussion in the manuscript and the comparison that the authors do (line 9-11, page 22) is hard to see from the figure.
- Page 21, line 16-18: Then why not excluding these from the analysis?

**Section 3.4**

This analysis should be changed. In order to fairly compare the effect of downscaling in the mean annual streamflow, which is what I assumed you are comparing (clarify), you should show results from the hydrological model calibrated using the non-downscaled precipitation (unless you did so, but it is not clear from the text) versus the calibrated hydrological model using downscaled precipitation. If you want to show this analysis you should also include the effect of downscaling on Nash-Sutcliffe, REHF and bias against observations.

---

## Referee Report (RR2)

Dear Editor and Authors,

Below is my review for the manuscript entitled: "Analysis of the streamflow extremes and long term water balance in Liguria Region of Italy using a cloud permitting grid spacing reanalysis dataset".

**General comments:**

The author's positively addressed most of my previous comments, however, there are some issues that I think remain in the manuscript and are detailed below. Main concerns are: (1) model parameters description, (2) forcing data used for calibration and (3) the discussion about attributing most - if not all - modelling issues to lack of calibration. I acknowledge that the writing improved, but I think the text can be further improved throughout the manuscript. I recommend moderate revisions to this paper before accepted to HESS. I think with just a bit more work this paper can be greatly improve and hopefully published.

**Specific comments:**

**Material and Methods**

- P7,L6: I don't think that a 1000 km2 basin can be considered small

- P7,L14: what do you  mean by "rigid" winters?

- P7,L5: why did you chose CDF among other approaches, can you name pros and cons (support your decision)?

- P7,L24: "seamlessly" I'm not sure if this is the adverb you want to use. Do you mean "continuously"?

- P8,L10: Add units.

- **There is no information about the parameters that requires no calibration, how were they obtained? For example, where is the Leaf Area Index (LAI) parameter coming from? Same for all the other parameters that are not mentioned in the manuscript (these are as important as calibrated parameters). This needs to be included.**

- P11,L16: Incoming shortwave?

- Still unclear the use of the term "section" in the manuscript, if you mean gauge station, sub-basin or basins, please use that instead, it is confusing.

- **I don't understand why the authors perform model calibration and validation using observed meteorological records, but then they move forward and do the analysis using the reanalysis product. In my opinion this is flawed and wrong, you should use the reanalysis to perform calibration/validation to prove that the model represent reasonably well the streamflow regime, and then, you can argue that the model is appropriate for the long-term analysis using reanalysis. Authors should reconsider this. If the reanalysis is really close to the observed meteorology (after all the corrections), results will show a good streamflow representation, but you need you show this.**

- P13,L13: It is unclear why you introduce Curve Number here? If its a parameters that the model requires it should be in a section describing the model parameters (with all the other parameters as well), otherwise explain.

- P14,L3-7: reword.

**Results**

- P19,L20-23: The authors blame calibration for all the mismatches they found. This is only one part of the problem. See comment below.

- **P20: 10-16 (figure 11): I think this is also the result of errors compensating and the authors should at least mention it (see the previously shown mismatches in figures 8 to 10 and table 5, to support the errors compensating). I can see that this can be useful for ungauged basins, but you have to keep in mind that there are a lot of uncertainties here that are not properly addressed in my opinion (not everything can be attributed to errors in parameter calibration, there are also uncertainties in model structure, non-calibrated parameters and input data). You should include a paragraph with a more comprehensive assessment (discussion) of the uncertainties and problems with the model. There is a lot of literature about this; here are some examples that you could look at:**

  - Liu and Gupta, 2007: Uncertainty in hydrologic modeling: Toward an integrated data assimilation framework, Water Resources Research, 10.1029/2006WR005756.
  - Wagener and Gupta, 2005: Model identification for hydrological forecasting under uncertainty: Stochastic Environmental Research and Risk Assessment, 10.1007/s00477-005-0006-5.
  - Walker et al., 2003: Defining Uncertainty: A Conceptual Basis for Uncertainty Management in Model-Based Decision Support, Integrated Assessment, 10.1076/iaij.4.1.5.16466.
  - Beven, K., 2007: A manifesto for the equifinality thesis, Journal of Hydrology, 10.1016/j.jhydrol.2005.07.007.

- Explicit the value of $A_{th}$ in page 20 and figure 11.

- P21,L15: This potential relationship between Ratio (T) and area is biased because the size of the sample is biased too (only few large basins and many small ones) and it should be stated in the text (not only here, but in other sections in the manuscript).

- Include a table with the details about the observed streamflow and meteorological data used in the study (period, gaps and official ID – if available-). This table should be included as supplementary material.

**Other minor comments:**

- Figure1: Add scale. Avoid acronyms (everywhere in the text too, unless previously defined). Such as FR, MC and IT. There is no curve number in the figure (see caption).

- Figure3: group the subplot in boxes by season and add an identifier, such as (a) for summer, etc. and describe it in the caption.

- Figure4: can probably be removed as the numbers are in Table 4 already (too many figures).

- Y-axe label figure 5 and 6: change to "**mean** monthly cumulative rainfall"

- Include legend in figure 5 and 6.

- Include axis units in figure 2, 3, 5, 6 and 16.

- Figure 8, 9 and 10: Change units format to "$m^3\ s^{-1}$".

- I still think that 16 figures is too much and some figures can be either merged, simplified or moved to supplement material.

- Table1: Include official ID number for the sub-basins (sections?). Don't capitalize "Slope". Typo in "height".

- Table2: What is "AP". Are these values for calibration or validation? Unclear. Include the period of analysis in the caption.

- Table5: What are the first "p-values" and "K-S test" associated with? Avoid using B.C. unless defined in the table caption.

- Table 6: What's "N. Program". Too many acronyms not defined throughout the text, avoid them unless explicitly defined.

- Abstract: What's "inter alia" streamflow?

- Many typos (more than previous version actually). Just to name a few examples:

  - Peri et al (2015) instead of Pieri et al (2015)
  - Krog et al (2015) instead of Krogh et al (2015)
  - Page7, L5: "et c" instead of "etc"
  - Page7, L9: there is a "." in the middle of the sentence.
  - Page 7, L22: ":" in the middle of the sentence.

---

## Author Response (AR2)

Dear reviewers and dear Editor,

We uploaded a new version of the paper. We inserted modifications answering to most of the reviewers requests. Various modifications were done along all the manuscript but we report the main structure changes:

- We made modifications on figures and added most of requested information
- Section devoted to hydrological model was re-organized in subsections and more details about the implementation were given
- The performance of the hydrological model used with mean parameters on calibrated basins were evaluated in order to evaluate the possible impact on ungauged basins.
- Analysis on annual maxima of accumulated rainfall on 24 hours was added.
- We removed references to later sections
- The language was revised by an expert of English language along all sections; if editor and reviewers agree that the paper is now improved regarding the scientific point of view but it still needs improvement regarding the language, we are open to submit the manuscript to a language revision service.

We hope the paper is now suitable for publication.

Answers to the reviewer comments are reproted in the following and we briefly described how we modified the manuscript.

The authors.

**Reviewer 1**

The reviewed manuscript corresponds to the authors response to the mayor revisions of the article: "Analysis of the streamflow extremes and long term water balance in Liguria Region of Italy using a cloud permitting grid spacing reanalysis dataset" by Silvestro et al.

**General comments**

- The authors present an approach (modelling chain) to use a long-term and high spatial resolution (4km) regional climate model with further bias correction to force a hydrological model over a large region in Italy. Calibration of the hydrological model was performed when possible and then parameters were transferred to ungauged basins. On gauged basins the model seems to represent relatively well streamflow; however, I see a significant limitation in the approach used to transfer these parameters to ungauged basins, as these are purely empirical, and the average of the calibrated parameters were used in ungauged. A GEV distribution was fitted to the simulated Annual Discharge Maxima (ADM) at all gauged sites and compared with observed streamflow, for which I see a fair agreement as fitted GEV 95% confidence interval sometimes does not cover observations (3 out of 6 in my opinion). I think the authors underestimate the role of the hydrological model in the overall analysis and discussion through the manuscript; more emphasis should be given to it, as this is largely driving the simulated hydrological response of these basins. The Ratio(t) is defined to assess the performance of a regionalization approach in characterizing streamflow for several return periods (T). Honestly I have somewhat a hard time understanding the usefulness of this regionalization approach, as it does not seems to work very well. I encourage the authors to demonstrate that the Ratio(T) is being well represented, as I see problems in small and large scale basins (Figure 11). The analysis of the effect of downscaling precipitation on the streamflow should be revised (see specific comments). The final analysis of the mass balance components (runoff ratio) seems very useful and maybe it should be more deeply explored and the main focus of the manuscript.

- I acknowledge that the authors responded to most of the comments from the previous revision; however, there are still significant issues with the manuscript in my opinion.

- For latter revisions I would encourage adding more details to the author's response letter, specifically where the changes are located in the manuscript (page and line), that way it would be much easier for us to go through the revisions.

- I believe that the English of the manuscript needs to be further improved.

**Specific Comments:**

**1 Introduction:**
- Page 4, Line 22-23: I agree in the use of high-resolution RCM to reproduce small-scale rainfall events, but I think you need a reference to this or even better if you can show that this is the case with the original dataset in your study region.
*Done, added four references (Buzzi et al., 2013; Marta-Almeida, 2016; Pontoppidan et al. 2017; Schwitalla et al. 2017).*
*Pgg 4, lines 19-20*

**2 Material and Methods:**
**Study area and case study:**
- Figure 1: To help the description of the figure, use (a), (b) and (c) to refer to each of the plot. And explain each of them as an inset of the others. I'm not sure about the utility of including the Curve Number, if you want to keep this you need to reference the source of this information. Also for the legend use "Curve Number" instead of C.N., try to avoid acronyms, same for DEM, etc.
*We changed the Figure as requested, CN map was removed*

- Page 5, line 12: refer to table 1.
*Ok done*
- Page 6, line 5: What about the raingage distribution vs elevation, is orographic precipitation well represented by these stations? Discuss.
*We added the fact that they are quite well distributed with elevation. In any case we used all available data.*
*Pgg 5, lines 18-22*
- Page 6, line 5-7: be more specific, how low?
*We added more details in the text: "…., respectively about 1/50, 1/200, 1/200, 1/60 km²"*
*Pgg 5, lines 22-25*

- Page 6, line 7-8: what data was used for calibration/validation?, I assume streamflow discharge, but it is unclear.
*-This is described some lines after those indicated by the reviewer (Page 6 lines 15-17 old text) and in the Hydrological model section. In any case we now used subsections in section 2.4, this should help in better clarify model description and its validation.*

*Pgg 12, lines 12-21*

**Downscaling the rainfall**
- Pag 10, line 10: Define Lr and tr**.**
*Done, we added a sentence to clarify "The model takes into account the variability of precipitation at small spatial and time scales (e.g. L ≤ 1 km, t ≤ 1 hour), preserving the precipitation volume at the scales considered reliable (Lr, and tr)  for quantitative precipitation forecasts. In other words Lr, and tr are those scales where we expect, on average, a reliable forecast of precipitation volume"*
*Pgg 9, lines 1-9*

- Pag 10, line 10-12: add reference.
*Done (Rebora et al., 2006a)*
*Pgg 9, lines 7-8*

**The hydrological model**
- This section needs better organization, use subsections if needed to explain: model inputs, calibration/validation and model structure (parameters, spatial resolution, etc.)
*The section is now re-organized in two subsections. The first describes generically the model the second focuses on the implementation for the current study-*
*Pgg 10 from line 10*

- How is evapotranspiration being calculated?

*We added a sentence and reference…"… The energy balance uses the "force restore equation" (Dickinson, 1988) that allows to explicitly model the soil surface temperature and estimating the evapotranspiration from the latent heat flux (Silvestro et. al 2013)…"*
*Pgg 11, lines 11-13*

- pag 13, line 15-17: Calibration procedure is not very clear. What do you mean by 11 sections, what's a "section" (basin?).

*These information are reported in Table 1, moreover we changed the text accordingly:".. The model was calibrated on 11 outlet sections where streamflow observations were available at hourly time resolution (see Table 1);"*
*Pgg 12, lines 12-13*

What ground stations measurements were interpolated? I assume you calibrate and validate the model using the downscalled reanalysis data, please clarify.

*No we used observations, we tried to clarify the text "…the hourly measurements of rainfall, air temperature, solar radiation, air relative humidity provided by the regional weather stations network were interpolated on a 1-km regular grid through a Kriging method fed to the model"*
*Pgg 12, lines 13-15*

- At what temporal scale did you calculated NS and REHF, hourly or daily?

*We added a sentence to clarify in section 2.4.2: "The observed streamflow data at 60- minute time resolution were compared with the model output in order to evaluate its performance. "*
*Pgg 12, lines 17-19*

- What range did you use for calibration and why?, explain. How did you come up with the 500 mm and 1 (Vxmax and Rf) for all the basins, wild guess?

*We added a table (Table 2 of new text) with the range of parameters and added a sentence in section 2.4.2 : "The parameters range values considered during the calibration process were defined considering their physical meaning, the mathematical constraints and the experience, they are reported in Table 2 (Silvestro et al., 2015; Cenci et al., 2016)"*
*Pgg 13, lines 15-17*

*Since Vxmax and Rf are less sensitive then the other 4 parameters we decided to fix them at regional scale similarly to what done in Davolio et al. (2017), we now highlight this fact in the text (section 2.4.2).*
*Pgg 12, lines 15-17*

- Pag. 14, line 16-17: This point is crucial in the manuscript. I disagree that taking average parameters values from calibration into ungauged basins is a satisfactory approach, especially as this are highly empirical values, and this is critical to analyze the results. I would encourage the authors to demonstrate that the impact of taking average parameters in ungauged basins is minimal. For this purpose you can use the average parameter values in the model of a gauged basin and demonstrate that these parameters have little impact on the streamflow performance. I think a significant assumption is being made here, which is not easy to support given the empirical basis of the model.

*We did the test requested by the reviewer. We added in table 3 (Table 2 old text) the statistics calculated on the model run by adopting average parameters also on calibrated sections. Clearly the performance are worse in respect using calibrated parameters, but as it can be clearly seen from table 3, the skill scores maintain good values. The new text highlights this finding.*
*Pgg 13,14 lines 21-2*

- For the regional analysis it is necessary to know how much of your study region is covered by gauged and ungauged basins, this can help discussing the impact of the assumptions in model's parameters.

*We added a sentence in section 2.6 "…The method was conceived and tested especially for the Tyrrhenian catchments of Liguria Region, so the present analysis was carried out only for this area; the 45-50% of which is located upstream the calibrated basins…"*
*Pgg 1, lines 19-22*

- Table 5: need to include the periods available for discharge, and ADM.
*As explicitly mention in the text (sections 2.6 and 3.5) data for ADM are not continuous and they are made of 20 to 50 years length time series often not continuous, the time windows of data availability are often not overlapped with ExpressHydro simulation. Similarly runoff ratio have been deducted by the official documents (we inserted also the link), which again are not continuous in time.*
*These are the reasons why we did not inserted the exact periods, but we explain the fact in the text.*
*Pgg 14,15 , lines 24-4*

**3 Results**
- Figure 2 and 3: I would recommend showing the difference in precipitation as a percentage with respect to the observed precipitation, or as (mm), so it is easier to compare with the other maps which are in mm.
*Done*

- Pag 17, line 17: not sure what you mean by "are largely confirmed"?. Please change the units of the difference map so the comparison is easier. And do the same when you analyze this difference.
*Ok, we changed the sentence: "..results on annual rainfall depth confirm the findings of Pieri et al. (2015) both on eastern and western Liguria sides"*

- Figure 4: Include coefficient of determination of correlation and mean bias.
*Done*

- Figure 5 and 6: Values presented in this figures are good for the analysis; however, I think you already have enough figures regarding the performance of EXPRESS-Hydro (Fig 2 to 6) I would try to merge or remove some, maybe by adding more information to Table 3 you can remove one or two plots without losing much information for the analysis.
*Figure 6 were added to satisfy the request of a reviewer during the first review, on the other side the regional perspective (Figure 5) is in our opinion interesting. We believe that they are both to be maintained, the total number of figures is high but similar to other published papers. In any case if editor and reviewer retain necessary remove one of them we will do it.*

- As ADM are a key aspect of your analysis, I would also show the performance of the bias-corrected precipitation time-series in representing peak precipitation, you could look at this by comparing the probability distribution functions of simulated and observed precipitation. I think this is very important as your peak flows are driven by the peak precipitation events, not by the precipitation volume. This will help you in the discussion of the ADM model's performance.
*Done, see pgg 18 lines 6 to 15 and Figure 7.*

- Pag 18, line 21-24: why did you chose these 4 basins and nor others?
*We added a comment in section 3.1: "..the basins locations was spread from east to west side of the region to investigate if different behaviours arise along the study area.."*
*Pgg 17, lines 23-24*

- You should try to merge figure 7, 8 and 9.
*We believe that it is better to maintain different figures in order to have more flexibility during editing (we have other big multi-panel pictures, as figure 3..); moreover we did some attempts to make a unique figure but we believe it results poorly readable and makes difficult to analyze the results*
*If editor and reviewer retain this point necessary we can do it, but we do not believe this improves the presentation of our work .*

- Why did you choose these 4 basins to contrast the analysis?
*We added a comment in section 3.1: "..Six basins were chosen in order to evidence the variability of results, showing either good and poor performances"*
*Pgg 18, lines 18-20*

- Explain what is that you are testing with Kolmorogov-smirnov test, and this should be in your methods too. I know you are talking about ADM distribution but it should be clear from in your manuscript.
*In section 2.6 we added a sentence "…Moreover the comparison between observed and modeled ADM was also done using the Kolmogorov-Smirnov test with a 5% significance level, in order to verify if they belonged to the same distribution.."*
*Pgg 15, lines 12-14*
*In section 3.2 we changed a sentence in order to explicitly refer to ADM…" The Kolmogorov-Smirnov test with a 5% significance level on ADM was applied to all the selected stations and the corresponding results are summarized in Table 5"*
*Pgg 19, lines 11-12*

**Section 3.3**
- Figure 10: are you using data from every grid point in your model to construct this curve or selected basin outlet?, this need to be clarified in the text. Avoid shortening the words in the figure; you have enough room in the plot, change 'Calib' to calibration, 'simul' to simulations, etc. This applies to all plots/tables.
*We modified the figures 7 to 10 (now 8 to 11).*
*The caption already mentioned explicitly that we used all grid points. We slightly modified it to better clarify: ".. Bottom panels: results without and with rainfall bias correction using all the grid points with drainage area larger than Ath."*

- Can you comment or interpret the step-changes in the observed growth curve of figure 10 in terms of the dominating hydrological processes in the region, and why you don't see that in the simulated values?
*The change of slope is quite evident in observations and simulated peaks on calibrated sections, for values of Qpeak/Mean around 2; it was found also by Boni et al. (2008) and it is mainly due to presence of extreme events in the time series. When considering the simulated series on all the grid points the curve is smoothed because of the availability of a larger amount of samples ( 30 years of ADM for every grid point of the region). The fact of having a lot of time series causes also the increase of slope for Qpeak/Mean very high (>4-5): the most extreme flow events pass on various grid points with similar effects in terms of Qpeak/Mean. Still the "double-component" behavior of the observed growth curve of figure 10 is still present in the simulated curve.*

- It seems to me that the relatively better performance in the regional growth curve, both calibrated and total area, could be the results of errors compensation as figure 7 to 9 show that observed distribution of ADM lies outside of the confidence boundary in some cases (Bisagno, Magra, Argentina; 3 of 6). This needs more discussion in the manuscript. If this approach is meant to be used in ungauged basins, the model needs to prove that it works in the calibrated basins. This can be a major problem in the proposed modelling chain. If the authors can prove that the model in calibrated basins can represent ADM, this will help a lot in supporting the proposed modelling chain.
*We added some comments to discuss more this issue, but we partially disagree with this point. The regional approach to build the growth curve is indeed used to reduce errors on single basins (see section 3.2, some ADM fitting are good some are bad). This is also highlighted (as already shown on the paper) by the comparison of the regional curve built with all grid points with the one built with the calibrated basins only. Her is the new text…"…. It is important to highlight that regional approach allows to reduce the errors that can be found for single basins (Boni et al., 2007), and which are shown in section 3.3; on one side the normalization of each ADM series with its average reduce the effects of bias (due for example to a bad hydrological model calibration), on the other side the ADM time series of each outlet section (or grid point) is only a small sub-sample of the entire sample used to build the regional curve…:"*

*Pgg 21, lines 1-5*

*Text already present:"…. The curves that used only calibrated sections are really similar to the others, proving that the latter configuration enhances the robustness of the regional curve estimation without introducing evident errors…"*

*Moreover in section 3.2 we already presented some comments about the fact that single bad fitting does not means that the general hydro-climatology of the region is badly described:"…. We would like also to highlight the fact that simulated ADM distributions have often similar shapes to the observed ones and suffer of a sort of bias (for example Bisagno closed at La Presa, Figure 8), while in other cases the simulated ADM distribution is only partially out of the confidence intervals (example Argentina closed at Merelli, Figure 9). The average hydrologic regime on the study region could be only partially affected by the local bad fittings..."*

*Finally, as requested by the reviewer, in section 2.4.2 the scores for calibrated basins are calculated using average parameters in order to evaluate the expected errors of hydrological model on un-calibrated outlet sections*

- Figure 11 shows that small basins are underestimated (as discussed in the manuscript), and it also shows an overestimation for the Ratio(T) for large basins, how can the authors explain this?. Also I'm not convinced that the B.C. Ratio(T) shows an improvement in Figure(11), as small basins that used to have a good performance (near 1) now they are overestimating (Ratio>1). Overall I'm doubtful about the usefulness of this Ratio(T) as it does not seem to produce good results. I think the authors need to show that this index, which tries to show that the regional curve is suitable for all basins, works.

*Ratio(T) is a simple way to compare the results with the benchmark (1 perfect; <1 underestimation; >1 overestimation). Many comments can be found to discuss results of Figures (11, 12 and also 13, now 12,13 and 14): page 21 and part of page 22 were devoted to discuss results and explain the reasons of these results.*
*In the new text we also evidenced that B.C. seems to introduce overestimation on large basins "….B.C. introduce an overestimation on larger basins (A > 200-300 km2)", while in text is already discussed the overestimation of the central part of the region also for small basins. The further analysis described in pages 21-22 and figure 14 evidences that mean Ratio improve using BIAS correction for all T, with values that remain in most of the cases (Fig 12 and 13) in range [0.5 1.5] and maximum range [0.3 2]; this is not a foregone results.*
*It is finally important to highlight (this comment is present also in the text) that we are comparing results with a "benchmark" and not with observations. As a consequence, also the benchmark can be affected by uncertainties, thus it appear to us an interesting achievement the fact that Quantiles have the same order of magnitude and in many cases differences lower than 50%.*
*We would like to highlight again that we are not presenting a definitive operational methodology, but we are exploring the potentialities of the presented modeling chain to reproduce ADM in an environment made by small basins, in complex topography areas, especially testing the usage of a high resolution (cloud permitting) reanalysis. To our knowledge this is one of the first works of this kind done in the considered study area and we have not the claim to solve all the possible issues and problems in a unique work, or to define a definitive methodology that can be applied in any case.*

- Figure 12 has little discussion in the manuscript and the comparison that the authors do (line 9-11, page 22) is hard to see from the figure.
*We added a sentence to help in reading the legend .." Areas where modelling chain is really close to the benchmark are in green/light blue, whereas dark blue and purple point out where under or overestimation is high (absolute difference larger than 70-100%)...".*

*We would like to notice that discussion of figure 12 (now 13) continues after lines 9-11 of page 22 of old text, results of figure were discussed even in the new text (page 22). In page 23 comments continue using both figures 12 and 13, and then introducing figure 14.*

- Page 21, line 16-18: Then why not excluding these from the analysis?
*We think the reviewer refers to page 23 not 21 i.e..".... leads us to consider that the underestimation of quantiles for very small catchments (i.e. A < 30-50 km^2 ) is a structural problem of themodelling chain...". The answer is that this is a finding (or result...) of the analysis, there were no a priori reasons to exclude them from the analysis.*

**Section 3.4**
This analysis should be changed. In order to fairly compare the effect of downscaling in the mean annual streamflow, which is what I assumed you are comparing (clarify), you should show results from the hydrological model calibrated using the non-downscaled precipitation (unless you did so, but it is not clear from the text) versus the calibrated hydrological model using downscaled precipitation. If you want to show this analysis you should also include the effect of downscaling on Nash-Sutcliffe, REHF and bias against observations.

*As described in section 2.5 (now 2.4.2 since we have inserted two sub-sections) the model was calibrated-validated using recent hourly data (2013-2014 period used for validation). The calibrated model is then used for simulating the 1979-2008 period using as input ExpressHydro Reanalysis with and without rainfall downscaling. So we have not different calibrations, but a unique calibration done for a period where time series of meteo gauges and discharge data are available.*

*It is not possible calculating the statistics like Nash-Sutcliffe, REHF...etc.., for 1979-2008 period, for 2 reasons: 1) no observed discharge time series are available 2)The ExpressHydro Reanalysis can not reproduce the exact sequence and timing of the real events in period 1979-2008, but only a reasonable climatology of the area. Theoretically it could be also reproduce particular events but with very high errors in geolocation and timing and large uncertainty from a quantitative point of view. As a consequence even in the case we had 1979-2008 observed discharge time series they could not be directly compared with simulated ones (both with and without downscaling)*

*The presented analysis is done to highlight the effects of downscaling especially on small basins. It appear to us that the presented graph which presents Ratio versus Area helps to synthetically show the downscaling effect, and the reason why we used it. In fact when ratio is far from 1 (lower than 1) it means that downscaling has an important role especially for small catchment (as commented in the text). Furthermore the fact that ratio is always < 1 means that the rainfall downscaling correctly plays the role of enhancing the runoff formation (we explicitly inserted this point in the new text)*

*We also added in the text the description of terms of the equation 9 to clarify the meaning of RatioDS*

**Reviewer 2**

The article presents a potentially useful study, but there are many aspects that still need to be improved before considering publication. Though the revised version has been partially improved, there are still portions of the methodology and text that are not clear. The organization is messy, forcing the reader to

move back and forth across sections.

Some of this issues and concerns were included in the previous reviews and unfortunately the authors were not able to address them in the revised manuscript.

For example, reviewer 1 of the previous round suggested moving the EXPRESS-Hydro reanalysis section to the introduction, as the text in lines 1 to 19 in page 7 are not part of the methodology. This was not done, and this section is still unclearly linked to the methodology (and this text belongs to the introduction)

*Ok. We already moved large part of this section on introduction, we now moved some other sentences and eliminated the subsection 2.2 (Express Hydro dataset)*

. He also suggested, replacing the Pieri et al., data by the Express-Hydro reanalysis everywhere in the text. This was not done everywhere, see for example line 9 in page 7, and later line 15 of page 17.... This is just one example of a problem, repeated in other aspects, that makes the manuscript (and work) very difficult to follow.

*Ok. Reference on page 7 has been removed. On section 3.1 we replaced Pieri et al.,with data by the Express-Hydro reanalysis when referring to the data set; we maintained the reference to Pieri et al. only when we refer to their results since we made similar analysis (comparison between ExpressHydro reanalysis and rainfall observations) but using a different set of observations. "…Pieri et al. (2015), using EURO4M-APGD reference observational dataset (Isotta et al. 2013, with about 50 daily raingauge stations over Liguria), already showed an overall underestimation of the WRF rainfall depths on annual basis in Liguria,..."...." The same analysis was repeated in this study, using 95 raingauge stations over Liguria.."*

There are constant references to later sections which again, make the manuscript difficult to follow.

*We removed references to later sections and added some references to scientific works along the manuscripts.*

Page 13 (and 14): What do you mean by "it was possible to calibrate the model for 11 sections?" how are "sections" defined? Which are these sections? Please clarify. Please comment on the range of variability of the parameter values, particularly as you the use a mean value for the ungagged basins ("sections?"). It might be worth doing a sensitivity analysis (using the range of calibrated parameters) to investigate the possible impact of using mean values.

*We now explicitly refer in the text to basin, we also changed the sentence to clarify which sections "…The model was calibrated on 11 basins where streamflow observations are available at hourly time resolution (see Table 1)…"*

*Under the suggestion of the other reviewer we did the run on calibrated outlet sections using average parameters (scores are reported in table 2), this should help to evaluate the impact on ungauged basins due to the use of average parameters.*

*We did not carry out a sensitive analysis because it is out of the scope of the paper, but a detailed sensitive analysis of the model parameters is reported in Silvestro et al. (2013).*

Page 17, section 3.1 is unclear. Are you referring to the EXPRESS-HYDRO reanalysis data?. Please, be clear in the introduction about this dataset, and refer here (in the same terms) to your new results. Is the bias

correction considered here? (it seems so, from line 14 in page 18, but please mention it at the beginning).

*Yes we changed the sentence at the beginning of the section: "The comparison between EXPRESS-Hydro reanalysis and precipitation climatology over Liguria from observational data was undertaken at the annual, seasonal and monthly scales"*

*The comparison of precipitation (Express Hydro versus observation) was done before bias correction was applied, we changed the sentence (page 18 line 14) which refers to skil scores calculation. "…while in Table 3 we reported the values of two statistics: BIAS and the Root Mean Square Error (RMSE)."*

*Only the analysis of 24 hours accumulation annual rainfall maxima is done with Bias Corrected reanalysis as requested by reviewer 1. This is explicitly mentioned in the text.*

*Text of the section has been revised*

The authors should consider a thorough English revision, possibly by a colleague with good English skills to improve readability and English usage. Please note also that the manuscript is full of one-sentence paragraphs, which makes the paper hard to read.
In short, the lack of clarity is compromising the delivery of the message of the paper.

*All the manuscript has been revised by an expert of English language*

**references**

Silvestro, F., Gabellani, S., Delogu, F., Rudari, R., Boni, G.: Exploiting remote sensing land surface

temperature in distributed hydrological modelling: the example of the Continuum model. Hydrol. Earth

Syst. Sci., 17, 39-62, 2013. doi:10.5194/hess-17-39-2013.

---

## Author Response (AR3)

Dear reviewers and dear Editor,
We uploaded a new version of the paper. We inserted modifications answering to most of the reviewers requests
We hope the paper is now suitable for publication.
Answers to the reviewer comments are reported in the following and we briefly described how we modified the manuscript.
The authors.

Reviewer1
**Material and Methods**
- P7,L6: I don't think that a 1000 km2 basin can be considered small
*We changed the sentence:"... characterized by small /medium sized (drainage area in the range 10-1000 km2) and steep slope (10 - 20 %) basins"*
- P7,L14: what do you mean by "rigid" winters?
*We changed the sentence: ". Winter seasons are generally not very cold being in a Mediterranean environment but,...."*
- P7,L5: why did you chose CDF among other approaches, can you name pros and cons (support your decision)?
*Done, in the text, section 2.2, it is already explained why the CDF matching was chosen. We also added some more information as requested, we report part of the text:*
*"The described procedure allowed to obtain a 3-hourly maps dataset in which the model bias was eliminated by keeping the characteristics of the modelled output in terms of seasonality and inter-annual variability. Furthermore, the procedure allows to avoid alterations of possible temporal trends, at both full-domain and single-cell spatial scale.*
*The CDF approach allows maintaining the most possible quantity of information furnished by the observations, namely the distribution of the monthly cumulative rainfall, this is not possible with simpler methods (like the simple correction of the average). On the other side, the temporal structure of the rainfall events at sub-monthly scale is the one derived from the model, that can bring some kind of distortion with respect to the actual meteorology of the region; the latter constitute the main limitation of the methodology."*

- P7,L24: "seamlessly" I'm not sure if this is the adverb you want to use. Do you mean "continuously"?
*Changed Seamlessly with continuously*

- P8,L10: Add units.
*We added the unit in the sentence before the formula: "....of rainfall p (3-hours cumulate in mm).."*

- There is no information about the parameters that requires no calibration, how were they obtained? For example, where is the Leaf Area Index (LAI) parameter coming from? Same for all the other parameters that are not mentioned in the manuscript (these are as important as calibrated parameters). This needs to be included.
*We added an Appendix where we show values of parameters which are not involved in calibration.*

- P11,L16: Incoming shortwave?
*Corrected*

- Still unclear the use of the term "section" in the manuscript, if you mean gauge station, sub-basin or basins, please use that instead, it is confusing.
*Corrected with gauge station or sub-basins along the manuscript*

- I don't understand why the authors perform model calibration and
validation using observed meteorological records, but then they move
forward and do the analysis using the reanalysis product. In my
opinion this is flawed and wrong, you should use the reanalysis to
perform calibration/validation to prove that the model represent
reasonably well the streamflow regime, and then, you can argue that
the model is appropriate for the long-term analysis using reanalysis.
Authors should reconsider this. If the reanalysis is really close to the
observed meteorology (after all the corrections), results will show a
good streamflow representation, but you need you show this.

*We do not agree with the referee, we also explained why we performed the calibration with
observations in our previous review, we report it in the following. Synthetically: a) performing
calibration with observations is the best option because it is not affected by the errors of
reanalysis and there is consistency between input (rain, ...) and output (streamflow) b)
reanalysis reproduces a possible realistic meteorological history but timing e location (and also
magnitude) of the events are not detailed enough so they are inconsistent with observed
streamflow c) Continuous observed streamflow for periods comparable with reanalysis are not
available (reliable observed streamflow are generally available for recent years)*
*This is the old answer:*
*....It is not possible calculating the statistics like Nash-Sutcliffe, REHF...etc.., for 1979-2008
period, for 2 reasons: 1) no observed discharge time series are available 2)The ExpressHydro
Reanalysis can not reproduce the exact sequence and timing of the real events in period 1979-
2008, but only a reasonable climatology of the area. Theoretically it could be also reproduce
particular events but with very high errors in geolocation and timing and large uncertainty from
a quantitative point of view. As a consequence even in the case we had 1979-2008 observed
discharge time series they could not be directly compared with simulated ones (both with and
without downscaling)*

*We added a paragraph in section 2.4.2 to explain the reasons of adopted approach.*
*"Model calibration was performed using observed input and output data without considering
reanalysis. This is in the authors opinion that best approach for several reasons: i) EXPRESS-
HYDRO reanalysis data are not involved because they are affected by errors larger than
observations ones ii) meteorological data at high time resolution are available iii) EXPRESS-
HYDRO reanalysis are too uncertain in terms of geo-location, timing and magnitude of real
events, as a consequence streamflow simulations can not be compared with observations iv) no
reliable streamflow data are available for the period covered by the reanalysis."*
*"*

- P13,L13: It is unclear why you introduce Curve Number here? If its a
parameters that the model requires it should be in a section describing the
model parameters (with all the other parameters as well), otherwise
explain.
*We added a sentence also in section 2.4.1:".. Curve Number maps are used to estimate some of
the subsurface flow parameters (Gabellani et al., 2008)...".*
*"*

- P14,L3-7: reword.

*Reworded: "Even if the calibration of the model did not encompass all the study region due to the lack of data, the hydrological model (as well) was proven to give good performances as well as similar models specifically developed for the same study area (Giannoni et al., 2000, 2005; Gabellani et al., 2008; Silvestro et al. 2013, 2015; Cenci et al 2016), even in not calibrated basins (Regione Marche, 2016)"*

Results
- P19,L20-23: The authors blame calibration for all the mismatches they found. This is only one part of the problem. See comment below.
- P20: 10-16 (figure 11): I think this is also the result of errors compensating and the authors should at least mention it (see the previously shown mismatches in figures 8 to 10 and table 5, to support the errors compensating). I can see that this can be useful for ungauged basins, but you have to keep in mind that there are a lot of uncertainties here that are not properly addressed in my opinion (not everything can be attributed to errors in parameter calibration, there are also uncertainties in model structure, non-calibrated parameters and input data). You should include a paragraph with a more comprehensive assessment (discussion) of the uncertainties and problems with the model. There is a lot of literature about this; here are some examples that you could look at:

Liu and Gupta, 2007: Uncertainty in hydrologic modeling: Toward an integrated data assimilation framework, Water Resources Research, 10.1029/2006WR005756.

Wagener and Gupta, 2005: Model identification for hydrological forecasting under uncertainty: Stochastic Environmental Research and Risk Assessment, 10.1007/s00477-005-0006-5.

Walker et al., 2003: Defining Uncertainty: A Conceptual Basis for Uncertainty Management in Model-Based Decision Support, Integrated Assessment, 10.1076/iaij.4.1.5.16466.

Beven, K., 2007: A manifesto for the equifinality thesis, Journal of Hydrology, 10.1016/j.jhydrol.2005.07.007.

*We added a paragraph in section 3.2 discussing the point of various sources of uncertainty (different from calibration) "....There are other sources of uncertainty that can explain the mismatches, hydrological model can lead to errors (Wlaker et al., 2003; Liu and Gupta, 2007) due to both its internal structure and to those parameters which are not calibrated but set by literature or by territorial information; furthermore the input variables are affected by uncertainty, not only rainfall, so they can be cause of errors."*

*We added a paragraph in section 3.3 to discuss the point of error compensation:"...It is anyway important to highlight that good results in terms of growth curve could be also partially due to the effect of error compensation..."*

*In the results chapter in different parts a deep discussion is done, we know that calibration is not the only source of uncertainty. The manuscript mentions the problem of meteorological input and also the problem of spatial and time scale of the precipitation, moreover the possible effect of bias correction.*

- Explicit the value of $A_{th}$ in page 20 and figure 11.
*Done, inserted value in the caption*

- P21,L15: This potential relationship between Ratio (T) and area is biased because the size of the sample is biased too (only few large basins and many small ones) and it should be stated in the text (not only here, but in other sections in the manuscript).
*In section 3.3 we highlighted this point. "The first consideration is that a relation between Ratio(T) and A appears to exist even if it is supposedly biased because the size of the sample is biased: the number of small basins is hugely higher than large ones"*

- Include a table with the details about the observed streamflow and meteorological data used in the study (period, gaps and official ID – if available-). This table should be included as supplementary material.
*We included a table with periods of availability of annual discharge maxima data. The other meteorological input (data for calibration and reanalysis ) are continuous and periods are already mentioned in the text.*
*The table should be included in supplementary materials*

**Other minor comments:**
- Figure1: Add scale. Avoid acronyms (everywhere in the text too, unless previously defined). Such as FR, MC and IT. There is no curve number in the figure (see caption).
*Done*

- Figure3: group the subplot in boxes by season and add an identifier, such as (a) for summer, etc. and describe it in the caption.
*Done*

- Figure4: can probably be removed as the numbers are in Table 4 already (too many figures).
*This figure was requested by a previous reviewer, it is in our opinion interesting to a visual check of results. If both editor and reviewer retain it must be removed we are open to do that.*

- Y-axe label figure 5 and 6: change to "mean monthly cumulative rainfall"
- Include legend in figure 5 and 6.
*Done*
- Include axis units in figure 2, 3(ANTO), 5, 6 and 16.
*Done*
- Figure 8, 9 and 10: Change units format to "m3 s-1".
*Done*
- I still think that 16 figures is too much and some figures can be either merged, simplified or moved to supplement material.
*The number of figures is quite high but the analysis is wide involving both rainfall and discharge. We would maintain the proposed figures to maintain a good readability, and even because they are derived after various reviews and merging process.*

- Table1: Include official ID number for the sub-basins (sections?). Don't capitalize "Slope". Typo in "height".
*Corrected. We do not have specific ID*

- Table2: What is "AP". Are these values for calibration or validation? Unclear.

Include the period of analysis in the caption.
*Done*
*CP: calibrated parameters AP: average parameters. We added the information in the caption*

- Table5: What are the first "p-values" and "K-S test" associated with? Avoid using B.C. unless defined in the table caption.
*We inserted the definition and modified the table. Results are reported for the two cases: with (B.C.) and without (No B.C.) rainfall bias correction.*

- Table 6: What's "N. Program". Too many acronyms not defined throughout the text, avoid them unless explicitly defined.
*Removed*

- Abstract: What's "inter alia" streamflow?
*Removed and changed the sentence*

- Many typos (more than previous version actually). Just to name a few examples:
o Peri et al (2015) instead of Pieri et al (2015)
o Krog et al (2015) instead of Krogh et al (2015)
o Page7, L5: "et c" instead of "etc"
o Page7, L9: there is a "." in the middle of the sentence.
o Page 7, L22: ":" in the middle of the sentence.
*Corrected*

[revised manuscript text omitted]